# Will Bilevel Optimizers Benefit from Loops

**Kaiyi Ji**
Department of CSE
University at Buffalo
kaiyiji@buffalo.edu

**Mingrui Liu**
Department of CS
George Mason University
mingruil@gmu.edu

**Yingbin Liang**
Department of ECE
The Ohio State University
liang.889@osu.edu

**Lei Ying**
Department of EECS
University of Michigan
leiying@umich.edu

## Abstract

Bilevel optimization has arisen as a powerful tool for solving a variety of machine learning problems. Two current popular bilevel optimizers AID-BiO and ITD-BiO naturally involve solving one or two sub-problems, and consequently, whether we solve these problems with loops (that take many iterations) or without loops (that take only a few iterations) can significantly affect the overall computational efficiency. Existing studies in the literature cover only some of those implementation choices, and the complexity bounds available are not refined enough to enable rigorous comparison among different implementations. In this paper, we first establish unified convergence analysis for both AID-BiO and ITD-BiO that are applicable to all implementation choices of loops. We then specialize our results to characterize the computational complexity for all implementations, which enable an explicit comparison among them. Our result indicates that for AID-BiO, the loop for estimating the optimal point of the inner function is beneficial for overall efficiency, although it causes higher complexity for each update step, and the loop for approximating the outer-level Hessian-inverse-vector product reduces the gradient complexity. For ITD-BiO, the two loops always coexist, and our convergence upper and lower bounds show that such loops are necessary to guarantee a vanishing convergence error, whereas the no-loop scheme suffers from an unavoidable non-vanishing convergence error. Our numerical experiments further corroborate our theoretical results.

## 1 Introduction

Bilevel optimization has attracted significant attention recently due to its popularity in a variety of machine learning applications including meta-learning [9, 1, 34, 17], hyperparameter optimization [9, 35, 5], reinforcement learning [22, 15], and signal processing [23, 7]. In this paper, we consider the bilevel optimization problem that takes the following formulation.

$$\min_{x \in \mathbb{R}^p} \Phi(x) := f(x, y^*(x)) \quad \text{s.t.} \quad y^*(x) = \arg\min_{y \in \mathbb{R}^q} g(x, y), \tag{1}$$

where the outer- and inner-level functions $f$ and $g$ are both jointly continuously differentiable. We focus on the setting where the lower-level function $g$ is strongly convex with respect to (w.r.t.) $y$ with the condition number $\kappa = \frac{L}{\mu}$ (where $L$ and $\mu$ are gradient Lipschitzness and strong convexity coefficients defined respectively in Assumptions 1 and 3 in Section 3), and the outer-level objective function $\Phi(x)$ is possibly nonconvex w.r.t. $x$. Such types of geometries arise in many applications

36th Conference on Neural Information Processing Systems (NeurIPS 2022).

Table 1: Comparison of computational complexities of four AID-BiO implementations for finding an $\epsilon$-accurate stationary point. For a fair comparison, gradient descent (GD) is used to solve the linear system for all algorithms. MV$(\epsilon)$: the total number of Jacobian- and Hessian-vector product computations. Gc$(\epsilon)$: the total number of gradient computations. $\widetilde{\mathcal{O}}$: hide $\ln \frac{\kappa}{\epsilon}$ factors. We write $a(x) = \Theta(b(x))$ if $cb(x) < a(x) < Cb(x)$, where $c, C$ are universal constants.

| Algorithms | $Q$ | $N$ | MV$(\epsilon)$ | Gc$(\epsilon)$ |
|---|---|---|---|---|
| BA [10] | $\Theta(\kappa \ln \kappa)$ | $\frac{(k+1)^{\frac{1}{4}}}{2}$ ($k$: iteration number) | $\widetilde{\mathcal{O}}(\kappa^5 \epsilon^{-1})$ | $\widetilde{\mathcal{O}}(\kappa^5 \epsilon^{-1.25})$ |
| AID-BiO [19] | $\Theta(\kappa \ln \kappa)$ | $\Theta(\kappa \ln \kappa)$ | $\widetilde{\mathcal{O}}(\kappa^4 \epsilon^{-1})$ | $\widetilde{\mathcal{O}}(\kappa^4 \epsilon^{-1})$ |
| $N$-$Q$-loop AID (this paper) | $\Theta(\kappa \ln \kappa)$ | $\Theta(\kappa \ln \kappa)$ | $\widetilde{\mathcal{O}}(\kappa^4 \epsilon^{-1})$ | $\widetilde{\mathcal{O}}(\kappa^4 \epsilon^{-1})$ |
| $Q$-loop AID (this paper) | $\Theta(\kappa \ln \kappa)$ | 1 | $\widetilde{\mathcal{O}}(\kappa^6 \epsilon^{-1})$ | $\widetilde{\mathcal{O}}(\kappa^5 \epsilon^{-1})$ |
| $N$-loop AID (this paper) | $\mathcal{O}(1)$ | $\Theta(\kappa \ln \kappa)$ | $\widetilde{\mathcal{O}}(\kappa^4 \epsilon^{-1})$ | $\widetilde{\mathcal{O}}(\kappa^5 \epsilon^{-1})$ |
| No-loop AID (this paper) | $\mathcal{O}(1)$ | 1 | $\widetilde{\mathcal{O}}(\kappa^6 \epsilon^{-1})$ | $\widetilde{\mathcal{O}}(\kappa^6 \epsilon^{-1})$ |

including meta-learning (which uses the last layer of neural networks as adaptation parameters), hyperparameter optimization (e.g., data hyper-cleaning and regularized logistic regression) and learning in communication networks (e.g., network utility maximization).

A variety of algorithms have been proposed to solve the bilevel optimization problem in eq. (1). For example, [14, 36, 32] proposed constraint-based approaches by replacing the inner-level problem with its optimality conditions as constraints. In comparison, gradient-based bilevel algorithms have received intensive attention recently due to the effectiveness and simplicity, which include two popular approaches via approximate implicit differentiation (AID) [4, 33, 11, 19] and iterative differentiation (ITD) [31, 8, 35]. Readers can refer to Appendix A for an expanded list of related work.

Consider the AID-based bilevel approach (which we call AID-BiO). Its base iteration loop updates the variable $x$ until convergence. Within such a base loop, it needs to solve two sub-problems: finding a nearly optimal solution of the inner-level function via $N$ iterations, and approximating the outer-level Hessian-inverse-vector product via $Q$ iterations. If $Q$ and $N$ are chosen to be large, then the corresponding iterations form **additional loops** of iterations within the base loop, which we respectively call as $Q$-loop and $N$-loop. Thus, AID-BiO can have four popular implementations depending on different choices of $N$ and $Q$: $N$-loop (with large $N = \kappa \ln \kappa$ and small $Q = \mathcal{O}(1)$), $N$-$Q$-loop (with large $N = \Theta(\kappa \ln \kappa)$ and large $Q = \Theta(\kappa \ln \kappa)$), $Q$-loop (with $N = 1$ and $Q = \Theta(\kappa \ln \kappa)$), and No-loop (with $N = 1$ and $Q = \mathcal{O}(1)$). Note that No-loop refers to no additional loops within the base loop, and can be understood as conventional single-(base)-loop algorithms. These implementations can significantly affect the efficiency of AID-BiO. Generally, large $Q$ (i.e., a $Q$-loop) provides a good approximation of the Hessian-inverse-vector product for the hypergradient computation, and large $N$ (i.e., a $N$-loop) finds an accurate optimal point of the inner function. Hence, an algorithm with $N$-loop and $Q$-loop require fewer base-loop steps to converge, but each such base-loop step requires more computations due to these loops. On the other hand, small $Q$ and/or $N$ avoid computations of loops in each base-loop step, but can cause the algorithm to converge with many more base-loop steps. An intriguing question here is which implementation is overall most efficient and whether AID-BiO benefits from having $N$-loop and/or Q-loop. Existing theoretical studies on AID-BiO are far from answering this question. The studies [10, 19] on deterministic AID-BiO focused only on the $N$-$Q$-loop scheme. A few studies analyzed the stochastic AID-BiO, such as [26] on No-loop, and [15, 21] on $Q$-loop. Those studies were not refined enough to capture the computational differences among different implementations, and further those studies collectively did not cover all the four implementations either.

- The first contribution of this paper lies in the development of a unified convergence theory for AID-BiO, which is applicable to all choices of $N$ and $Q$. We further specialize our general theorems to provide the computational complexity for all of the above four implementations (as summarized in Table 1). Comparison among them suggests that AID-BiO does benefit from both $N$-loop and $Q$-loop. This is in contrast to minimax optimization (a special case of bilevel optimization), where it is shown in [27, 41] that (No-loop) gradient descent ascent (GDA) with $N = 1$ often outperforms ($N$-loop) GDA with $N = \kappa \ln \kappa$ (here $N$ denotes the number of ascent iterations for each descent iteration). To explain the reason, the gradient

Table 2: Comparison of computational complexities of two ITD-BiO implementations for finding an $\epsilon$-accurate stationary point. For a fair comparison, gradient descent (GD) is used to solve the inner-level problem. The analysis in [19] for ITD-BiO assumes that the inner-loop minimizer $y^*(x_k)$ is bounded at $k^{th}$ iteration, which is not required in our analysis. $\mu$: the strong-convexity constant of inner-level function $g(x, \cdot)$. For the last two columns, 'N/A' means that the complexities to achieve an $\epsilon$-accuracy are not measurable due to the nonvanishing convergence error. We write $a(x) = \Omega(b(x))$ if $a(x) > cb(x)$, where $c$ is a universal constant.

| **Algorithms** | $N$ | Convergence rate | **MV**$(\epsilon)$ | **Gc**$(\epsilon)$ |
|---|---|---|---|---|
| ITD-BiO [19] | $\Theta(\kappa \ln \kappa)$ | $\mathcal{O}\left(\frac{\kappa^3}{K} + \epsilon\right)$ | $\widetilde{\mathcal{O}}(\kappa^4 \epsilon^{-1})$ | $\widetilde{\mathcal{O}}(\kappa^4 \epsilon^{-1})$ |
| $N$-$N$-loop ITD (this paper) | $\Theta(\kappa \ln \kappa)$ | $\mathcal{O}\left(\frac{\kappa^3}{K} + \epsilon\right)$ | $\widetilde{\mathcal{O}}(\kappa^4 \epsilon^{-1})$ | $\widetilde{\mathcal{O}}(\kappa^4 \epsilon^{-1})$ |
| No-loop ITD (this paper) | $\Theta(1)$ | $\mathcal{O}\left(\frac{\kappa^3}{K} + \kappa^3\right)$ | N/A | N/A |
| Lower bound (this paper) | $\Theta(1)$ | $\Omega\left(\kappa^2\right)$ | N/A | N/A |

w.r.t. $x$ in bilevel optimization involves additional second-order derivatives (that do not exist in minimax optimization), which are more sensitive to the accuracy of the optimal point of the inner function. Therefore, a large $N$ finds such a more accurate solution, and is hence more beneficial for bilevel optimization than minimax optimization.

Differently from AID-BiO, the ITD-based bilevel approach (which we call as ITD-BiO) constructs the outer-level hypergradient estimation via backpropagation along the $N$-loop iteration path, and $Q = N$ always holds. Thus, ITD-BiO has only two implementation choices: $N$-$N$-loop (with large $N = \kappa \ln \kappa$) and No-loop (with small $N = \mathcal{O}(1)$). Here, $N$-$N$-loop and No-loop also refer to additional loops for solving sub-problems within the ITD-BiO's base loop of updating the variable $x$. The only convergence rate analysis on ITD-BiO was provided in [19] but only for $N$-$N$-loop, which does not suggest how $N$-$N$-loop compares with No-loop. It is still an open question whether ITD-BiO benefits from $N$-loops.

- The second contribution of this paper lies in the development of a unified convergence theory for ITD-BiO, which is applicable to all values of $N$. We then specialize our general theorem to provide the computational complexity for both of the above implementations (as summarized in Table 2). We further develop a convergence lower bound, which suggests that $N$-$N$-loop is necessary to guarantee a vanishing convergence error, whereas the no-loop scheme suffers from an unavoidable non-vanishing convergence error.

The technical contribution of this paper is two-fold. For AID methods, most existing studies including [19] solve the linear system with large $Q = \Theta(\kappa \log \kappa)$ so that the upper-level Hessian-inverse-vector product approximation error can vanish. In contrast, we allow arbitrary (possibly small) $Q$, and hence this upper-level error can be large and nondecreasing, posing a key challenge to guarantee convergence. We come up with a novel idea to prove the convergence by showing that this error, not by itself but jointly with the inner-loop error, admits an (approximately) iteratively decreasing property, which bounds the hypergradient error and yields convergence. The analysis contains new developments to handle the coupling between this error and the inner-loop error, which is critical in our proof. For ITD methods, unlike existing studies including [19], we remove the boundedness assumption on $y^*(x)$ via a novel error analysis over the entire execution rather than a single iteration. Our analysis tools are general and can be extended to stochastic and acceleration bilevel optimizers.

## 2  Algorithms

### 2.1  AID-based Bilevel Optimization Algorithm

As shown in Algorithm 1, we present the general AID-based bilevel optimizer (which we refer to AID-BiO for short). At each iteration $k$ of the base loop, AID-BiO first executes $N$ steps of gradient decent (GD) over the inner function $g(x, y)$ to find an approximation point $y_k^N$, where $N$ can be chosen either at a constant level or as large as $N = \kappa \ln \kappa$ (which forms an $N$-**loop** of iterations). Moreover, to accelerate the practical training and achieve a stronger performance guarantee, AID-BiO

---

**Algorithm 1** AID-based bilevel optimization (AID-BiO) with double warm starts

---

1: **Input:** Stepsizes $\alpha, \beta > 0$, initializations $x_0, y_0, v_0$.
2: **for** $k = 0, 1, 2, ..., K$ **do**
3:     Set $y_k^0 = y_{k-1}^N$ if $k > 0$ and $y_0$ otherwise *(warm start initialization)*
4:     **for** $t = 1, ...., N$ **do**
5:         Update $y_k^t = y_k^{t-1} - \alpha \nabla_y g(x_k, y_k^{t-1})$
6:     **end for**
7:     Hypergradient estimation via:
        Set $v_k^0 = v_{k-1}^Q$ if $k > 0$ and $v_0$ otherwise *(warm start initalization)*.
        Solve $v_k^Q$ from $\nabla_y^2 g(x_k, y_k^N) v = \nabla_y f(x_k, y_k^N)$ via $Q$ steps of iterative algorithms starting from $v_k^0$
        Compute $\widehat{\nabla} \Phi(x_k) = \nabla_x f(x_k, y_k^N) - \nabla_x \nabla_y g(x_k, y_k^N) v_k^Q$
8:     Update $x_{k+1} = x_k - \beta \widehat{\nabla} \Phi(x_k)$
9: **end for**

---

often adopts a warm-start strategy by setting the initialization $y_k^0$ of each $N$-loop to be the output $y_{k-1}^N$ of the preceding $N$-loop rather than a random start.

To update the outer variable, AID-BiO adopts the gradient descent, by approximating the true gradient $\nabla \Phi(x_k)$ of the outer function w.r.t. $x$ (called hypergradient [33, 11]) that takes the following form:

$$\text{(True hypergradient:)} \quad \nabla \Phi(x_k) = \nabla_x f(x_k, y^*(x_k)) - \nabla_x \nabla_y g(x_k, y^*(x_k)) v_k^*, \tag{2}$$

where $v_k^*$ is the solution of the linear system $\nabla_y^2 g(x_k, y^*(x_k)) v = \nabla_y f(x_k, y^*(x_k))$. To approximate the above true hypergradient, AID-BiO first solves $v_k^Q$ as an approximate solution to a linear system $\nabla_y^2 g(x_k, y_k^N) v = \nabla_y f(x_k, y_k^N)$ using $Q$ steps of GD iterations starting from $v_k^0$. Here, $Q$ can also be chosen either at a constant level or as large as $Q = \kappa \ln \frac{\kappa}{\mu}$ (which forms a $Q$-**loop** of iterations). Note that a warm start is also adopted here by setting $v_k^0 = v_{k-1}^Q$, which is critical to achieve the convergence guarantee for small $Q$. If $Q$ is large enough, e.g., at an order of $\kappa \ln \frac{\kappa}{\mu}$, a zero initialization with $v_k^0 = 0$ suffices to solve the linear system well. Then, AID-BiO constructs a hypergradient estimator $\widehat{\nabla} \Phi(x_k)$ given by

$$\text{(AID-based hypergradient estimate:)} \quad \widehat{\nabla} \Phi(x_k) = \nabla_x f(x_k, y_k^N) - \nabla_x \nabla_y g(x_k, y_k^N) v_k^Q. \tag{3}$$

Note that the execution of AID-BiO involves only Hessian-vector products in solving the linear system and Jacobian-vector product $\nabla_x \nabla_y g(x_k, y_k^N) v_k^Q$ which are more computationally tractable than the calculation of second-order derivatives.

It is clear that different choices of $N$ and $Q$ lead to four implementations within the base loop of AID-BiO: $N$-loop (with large $N = \kappa \ln \kappa$ and small $Q = \mathcal{O}(1)$), $N$-$Q$-loop (with large $N = \kappa \ln \kappa$ and $Q = \kappa \ln \kappa$), $Q$-loop (with small $N = 1$ and large $Q = \kappa \ln \kappa$) and No-loop (with small $N = 1$ and $Q = \mathcal{O}(1)$). In Section 4, we will establish a unified convergence theory for AID-BiO applicable to all its implementations in order to formally compare their computational efficiency.

## 2.2 ITD-Based Bilevel Optimization Algorithm

As shown in Algorithm 2, the ITD-based bilevel optimizer (which we refer to as ITD-BiO) updates the inner variable $y$ similarly to AID-BiO, and obtains the $N$-step output $y_k^N$ of GD with a warm-start initialization. ITD-BiO differentiates from AID-BiO mainly in its estimation of the hypergradient. Without leveraging the implicit gradient formulation, ITD-BiO computes a direct derivative $\frac{\partial f(x_k, y_k^N)}{\partial x_k}$ via automatic differentiation for hypergradient approximation. Since $y_k^N$ has a dependence on $x_k$ through the $N$-loop iterative GD updates, the execution of ITD-BiO takes the backpropagation over the entire $N$-loop trajectory. To elaborate, it can be shown via the chain rule that the hypergradient estimate $\frac{\partial f(x_k, y_k^N)}{\partial x_k}$ takes the following form of $\frac{\partial f(x_k, y_k^N)}{\partial x_k} = \nabla_x f(x_k, y_k^N) - \alpha \sum_{t=0}^{N-1} \nabla_x \nabla_y g(x_k, y_k^t) \prod_{j=t+1}^{N-1} (I - \alpha \nabla_y^2 g(x_k, y_k^j)) \nabla_y f(x_k, y_k^N)$. As shown in this equation, the differentiation does not compute the second-order derivatives directly but compute more tractable and economical Hessian-vector products $\nabla_y^2 g(x_k, y_k^{j-1}) v_j, j = 1, ..., N$

---

**Algorithm 2** ITD-based bilevel optimization algorithm (ITD-BiO) with warm start

---
1: **Input:** Stepsize $\alpha > 0$, initializations $x_0$ and $y_0$ .
2: **for** $k = 0, 1, 2, ..., K$ **do**
3:    Set $y_k^0 = y_{k-1}^N$ if $k > 0$ and $y_0$ otherwise *(warm start initialization)*
4:    **for** $t = 1, ...., N$ **do**
5:       Update $y_k^t = y_k^{t-1} - \alpha \nabla_y g(x_k, y_k^{t-1})$
6:    **end for**
7:    Compute $\widehat{\nabla}\Phi(x_k) = \frac{\partial f(x_k, y_k^N)}{x_k}$ via backpropagation w.r.t. $x_k$
8:    Update $x_{k+1} = x_k - \beta \widehat{\nabla}\Phi(x_k)$
9: **end for**

---

(similarly for Jacobian-vector products), where each $v_j$ is obtained recursively via $v_{j-1} = (I - \alpha \nabla_y^2 g(x_m, y_m^j)) v_j$ with $v_N = \nabla_y f(x_m, y_m^N)$.

Clearly, the implementation of ITD-BiO implies that $N = Q$ always holds. Hence, ITD-BiO takes only two possible architectures within its base loop: $N$-$N$-loop (with large $N = \kappa \ln \frac{\kappa}{\epsilon}$) and No-loop (with small $N = 1$). In Section 5, we will establish a unified convergence theory for ITD-BiO applicable to both of its implementations in order to formally compare their computational efficiency.

## 3 Definitions and Assumptions

This paper focuses on the following types of objective functions.

**Assumption 1.** *The inner-level function $g(x, y)$ is $\mu$-strongly-convex w.r.t. $y$.*

Since the objective function $\Phi(x)$ in eq. (1) is possibly nonconvex, algorithms are expected to find an $\epsilon$-accurate stationary point defined as follows.

**Definition 1.** *We say $\bar{x}$ is an $\epsilon$-accurate stationary point for the bilevel optimization problem given in eq. (1) if $\|\nabla\Phi(\bar{x})\|^2 \leq \epsilon$, where $\bar{x}$ is the output of an algorithm.*

In order to compare the performance of different bilevel algorithms, we adopt the following metrics of computational complexity.

**Definition 2.** *Let $\mathrm{Gc}(\epsilon)$ be the number of gradient evaluations, and $\mathrm{MV}(\epsilon)$ be the total number of Jacobian- and Hession-vector product evaluations to achieve an $\epsilon$-accurate stationary point of the bilevel optimization problem in eq. (1).*

Let $z = (x, y)$. We take the following standard assumptions, as also widely adopted by [10, 17].

**Assumption 2.** *Gradients $\nabla f(z)$ and $\nabla g(z)$ are L-Lipschitz, i.e., for any $z, z'$,*

$$\|\nabla f(z) - \nabla f(z')\| \leq L\|z - z'\|, \quad \|\nabla g(z) - \nabla g(z')\| \leq L\|z - z'\|.$$

As shown in eq. (2), the gradient of the objective function $\Phi(x)$ involves the second-order derivatives $\nabla_x \nabla_y g(z)$ and $\nabla_y^2 g(z)$. The following assumption imposes the Lipschitz conditions on such higher-order derivatives, as also made in [10].

**Assumption 3.** *Suppose the derivatives $\nabla_x \nabla_y g(z)$ and $\nabla_y^2 g(z)$ are $\rho$-Lipschitz, i.e., for any $z, z'$*

$$\|\nabla_x \nabla_y g(z) - \nabla_x \nabla_y g(z')\| \leq \rho\|z - z'\|, \quad \|\nabla_y^2 g(z) - \nabla_y^2 g(z')\| \leq \rho\|z - z'\|.$$

To guarantee the boundedness the hypergradient estimation error, existing works [10, 17, 11] assume that the gradient $\nabla f(z)$ is bounded for all $z = (x, y)$. Instead, we make a weaker boundedness assumption on the gradients $\nabla_y f(x, y^*(x))$.

**Assumption 4.** *There exists a constant $M$ such that for any $x$, $\|\nabla_y f(x, y^*(x))\| \leq M$.*

For the case where the total objective function $\Phi(\cdot)$ has some benign structures, e.g., convexity or strong convexity, Assumption 4 can be removed by an induction analysis that all iterates are bounded as in [18]. Assumption 4 can also be removed by projecting $x$ onto a bounded constraint set $\mathcal{X}$.

# 4 Convergence Analysis of AID-BiO

As we describe in Section 2.1, AID-BiO can have four possible implementations depending on whether $N$ and $Q$ are chosen to be large enough to form an $N$-loop and/or $Q$-loop. In this section, we will provide the convergence analysis and characterize the overall computational complexity for all of the four implementations, which will provide the general guidance on which algorithmic architecture is computationally most efficient.

## 4.1 Convergence Rate and Computational Complexity

In this subsection, we develop two unified theorems for AID-BiO, both of which are applicable to all the regimes of $N$ and $Q$. We then specialize these theorems to provide the complexity bounds (as corollaries) for the four implementations of AID-BiO. It turns out that the first theorem provides tighter complexity bounds for the implementations with small $Q = \Theta(1)$, and the second theorem provides tighter complexity bounds for the implementations with large $Q = \kappa \ln \frac{\kappa}{\epsilon}$. Our presentation of those corollaries below will thus focus only on the tighter bounds. The following theorem provides our first unified convergence analysis for AID-BiO.

**Theorem 1.** *Suppose Assumptions 1, 2, 3 and 4 hold. Choose parameters $\alpha, \eta$ and $\lambda$ such that $(1+\lambda)(1-\alpha\mu)^N(1+r(1+\frac{1}{\eta\mu})) \le 1 - \eta\mu$, where $r = \Theta(\mu^2 C_Q^2)$ with $C_Q = \Theta\big((1-\eta\mu)^{Q-1}\frac{\eta Q}{\mu} + \frac{1-(1-\eta\mu)^Q(1+\eta Q\mu)}{\mu^2} + (1-(1-\eta\mu)^Q)\frac{L}{\mu}\big)$. Let $L_\Phi = \Theta(\kappa^3)$ be the smoothness parameter of $\Phi(\cdot)$. Let $\widetilde{w} := \Theta\big(\frac{\eta\mu\kappa^4}{\lambda r} + \frac{\kappa^4}{\eta\mu}\big(\frac{(1-\eta\mu)^{2Q}}{\mu^2} + \frac{\eta\mu}{\lambda}\big)\big)$. Choose $\beta$ such that $\beta = \min\big\{\frac{1}{L_\Phi}, \sqrt{\frac{\eta\mu}{\widetilde{w}}}\big\}$. Then,*

$$\frac{1}{K}\sum_{k=0}^{K-1}\|\nabla\Phi(x_k)\|^2 = \mathcal{O}\Big(\frac{\Phi(x_0) - \Phi(x^*)}{\beta K} + \frac{\kappa^2\|y_0^*\|^2 + (\frac{3M}{\mu} + \kappa^2)}{\eta\mu K}\Big). \tag{4}$$

The complete version of Theorem 1 with full parameter specifications can be found in Appendix H. Theorem 1 also elaborates the precise requirements on the stepsizes $\alpha, \eta$ and $\beta$ and the auxiliary parameter $\lambda$, which take complicated forms. In the following, by further specifying these parameters, we characterize the complexities for AID-BiO in more explicit forms. We focus on the implementations with $Q = \Theta(1)$ (for which Theorem 1 specializes to tighter bound than Theorem 2 below), which includes the $N$-loop scheme (with $N = \Theta(\kappa \ln \kappa)$) and the No-loop scheme (with $N = 1$).

**Corollary 1** ($N$-loop). *Consider $N$-loop AID-BiO with $N = \Theta(\kappa \ln \kappa)$ and $Q = \Theta(1)$, where $\kappa = \frac{L}{\mu}$ denotes the condition number of the inner problem. Under the same setting of Theorem 1, choose $\eta = \frac{1}{L}$, $\alpha = \frac{1}{L}$, and $\lambda = 1$. Then, we have $\frac{1}{K}\sum_{k=0}^{K-1}\|\nabla\Phi(x_k)\|^2 = \mathcal{O}\big(\frac{\kappa^4}{K} + \frac{\kappa^3}{K}\big)$, and the complexity to achieve an $\epsilon$-accurate stationary point is $\mathrm{Gc}(\epsilon) = \widetilde{\mathcal{O}}(\kappa^5\epsilon^{-1}), \mathrm{MV}(\epsilon) = \widetilde{\mathcal{O}}\big(\kappa^4\epsilon^{-1}\big)$.*

**Corollary 2** (No-loop). *Consider No-loop AID-BiO with $N = 1$ and $Q = \Theta(1)$. Under the same setting of Theorem 1, choose parameters $\alpha = \frac{1}{L}$, $\lambda = \frac{\alpha\mu}{2}$ and $\eta = \min\{\frac{1}{128}\frac{\alpha\mu^2}{Q^2L^2}, \frac{\alpha}{4}, \frac{1}{\mu Q}\}$. Then, $\frac{1}{K}\sum_{k=0}^{K-1}\|\nabla\Phi(x_k)\|^2 = \mathcal{O}\big(\frac{\kappa^6}{K} + \frac{\kappa^5}{K}\big)$, and the complexity is $\mathrm{Gc}(\epsilon) = \widetilde{\mathcal{O}}(\kappa^6\epsilon^{-1}), \mathrm{MV}(\epsilon) = \widetilde{\mathcal{O}}(\kappa^6\epsilon^{-1})$.*

The analysis of Theorem 1 can be further improved for the large $Q$ regime, which guarantees a sufficiently small outer-level approximation error, and helps to relax the requirement on the stepsize $\eta$. Such an adaptation yields the following alternative unified convergence characterization for AID-BiO, which is applicable for all $Q$ and $N$, but specializes to tighter complexity bounds than Theorem 1 in the large $Q$ regime. For simplicity, we set the initialization $v_k^0 = 0$ in Algorithm 1.

**Theorem 2.** *Suppose Assumptions 1, 2, 3 and 4 hold. Define $\tau = \Theta\big((1-\alpha\mu)^N(1+\lambda+(1+\lambda^{-1})(\kappa^2+C_Q^2)\kappa^2\beta^2)\big)$, $w = \Theta\big((1-\alpha\mu)^N(\kappa^2+C_Q^2)(1+\lambda^{-1})\kappa^2\big)$, where $C_Q$ is a positive constant defined as in Theorem 1. Choose parameters $\alpha, \beta$ such that $\tau < 1$ and $\beta L_\Phi + w\beta^2\big(\frac{1}{2} + \beta L_\Phi\big)\frac{1}{1-\tau} \le \frac{1}{4}$ hold. Then, the output of AID-BiO satisfies*

$$\frac{1}{K}\sum_{k=0}^{K-1}\|\nabla\Phi(x_k)\|^2 = \mathcal{O}\Big(\frac{\Phi(x_0) - \Phi(x^*)}{\beta K} + \frac{1}{K}\frac{\delta_0}{1-\tau} + \kappa^2(1-\eta\mu)^{2Q}\Big),$$

*where $\delta_0 = \Theta((\kappa^2 + C_Q^2)(1-\alpha\mu)^N\|y_0^* - y_0\|^2)$ is the initial distance.*

The complete version of Theorem 2 with full parameter specifications can be found in Appendix K. We next specialize Theorem 2 to obtain the complexity for two implementations of AID-BiO with

$Q = \Theta(\kappa \ln \kappa)$: $N$-$Q$-loop (with $N = \Theta(\kappa \ln \kappa)$) and $Q$-loop (with $N = 1$), as shown in the following two corollaries. For each case, we need to set the parameters $\lambda, \eta$ and $\alpha$ in Theorem 2 properly.

**Corollary 3** ($N$-$Q$-loop). *Consider $N$-$Q$-loop AID-BiO with $N = \Theta(\kappa \ln \kappa)$ and $Q = \Theta(\kappa \ln \frac{\kappa}{\epsilon})$. Under the same setting of Theorem 2, choose $\eta = \alpha = \frac{1}{L}$, $\lambda = 1$ and $\beta = \Theta(\kappa^{-3})$. Then, $\frac{1}{K}\sum_{k=0}^{K-1} \|\nabla\Phi(x_k)\|^2 = \mathcal{O}\left(\frac{\kappa^3}{K} + \epsilon\right)$, and the complexity is $\mathrm{Gc}(\epsilon) = \widetilde{\mathcal{O}}(\kappa^4 \epsilon^{-1})$, $\mathrm{MV}(\epsilon) = \widetilde{\mathcal{O}}(\kappa^4 \epsilon^{-1})$.*

**Corollary 4** ($Q$-loop). *Consider $Q$-loop AID-BiO with $N = 1$ and $Q = \Theta(\kappa \ln \frac{\kappa}{\epsilon})$. Under the same setting of Theorem 2, choose $\alpha = \eta = \frac{1}{L}$, $\lambda = \frac{\alpha\mu}{2}$ and $\beta = \Theta(\kappa^{-4})$. Then, $\frac{1}{K}\sum_{k=0}^{K-1} \|\nabla\Phi(x_k)\|^2 = \mathcal{O}\left(\frac{\kappa^5}{K} + \frac{\kappa^4}{K} + \epsilon\right)$, and the complexity is $\mathrm{Gc}(\epsilon) = \widetilde{\mathcal{O}}(\kappa^5 \epsilon^{-1})$, $\mathrm{MV}(\epsilon) = \widetilde{\mathcal{O}}(\kappa^6 \epsilon^{-1})$.*

### 4.2 Comparison among Four Implementations

**Impact of $N$-loop ($N = 1$ vs $N = \kappa \ln \kappa$).** We fix $Q$, and compare how the choice of $N$ affects the computational complexity. First, let $Q = \Theta(1)$, and compare the results between the two implementations $N$-loop with $\Theta(\kappa \ln \kappa)$ (Corollary 1) and No-loop with $N = 1$ (Corollary 2). Clearly, the $N$-loop scheme significantly improves the convergence rate of the No-loop scheme from $\mathcal{O}\left(\frac{\kappa^6}{K}\right)$ to $\mathcal{O}\left(\frac{\kappa^4}{K}\right)$, and improves the matrix-vector and gradient complexities from $\widetilde{\mathcal{O}}(\kappa^6 \epsilon^{-1})$ and $\widetilde{\mathcal{O}}(\kappa^6 \epsilon^{-1})$ to $\widetilde{\mathcal{O}}(\kappa^4 \epsilon^{-1})$ and $\widetilde{\mathcal{O}}(\kappa^5 \epsilon^{-1})$, respectively. To explain intuitively, the hypergradient estimation involves a coupled error $\eta \|y_k^N - y^*(x_k)\|$ induced from solving the linear system $\nabla_y^2 g(x_k, y_k^N)v = \nabla_y f(x_k, y_k^N)$ with stepsize $\eta$. Therefore, a smaller inner-level approximation error $\|y_k^N - y^*(x_k)\|$ allows a more aggressive stepsize $\eta$, and hence yields a faster convergence rate as well as a lower total complexity, as also demonstrated in our experiments. It is worth noting that such a comparison is generally different from that in minimax optimization [27, 41], where alternative (i.e., No-loop) gradient descent ascent (GDA) with $N = 1$ outperforms (N-loop) GDA with $N = \kappa \ln \kappa$, where $N$ denotes the number of ascent iterations for each descent iteration. To explain the reason, in constrast to minimax optimization, the gradient w.r.t. $x$ in bilevel optimization involves **additional** second-order derivatives, which are more sensitive to the inner-level approximation error. Therefore, a larger $N$ is more beneficial for bilevel optimization than minimax optimization. Similarly, we can also fix $Q = \Theta(\kappa \ln \kappa)$, the $N$-$Q$-loop scheme with $N = \kappa \ln \kappa$ (Corollary 3) significantly outperforms the $Q$-loop scheme with $N = 1$ (Corollary 4) in terms of the convergence rate and complexity.

**Impact of $Q$-loop ($Q = 1$ vs $Q = \Theta(\kappa \ln \frac{\kappa}{\epsilon})$).** We fix $N$, and characterize the impact of the choice of $Q$ on the complexity. For $N = 1$, comparing No-loop with $Q = \Theta(1)$ in Corollary 2 and $Q$-loop with $Q = \Theta(\kappa \ln \kappa)$ in Corollary 4 shows that both choices of $Q$ yield the same matrix-vector complexity $\widetilde{\mathcal{O}}(\kappa^6 \epsilon^{-1})$, but $Q$-loop with a larger $Q$ improves the gradient complexity of No-loop with $Q = \Theta(1)$ from $\widetilde{\mathcal{O}}(\kappa^6 \epsilon^{-1})$ to $\widetilde{\mathcal{O}}(\kappa^5 \epsilon^{-1})$. A similar phenomenon can be observed for $N = \Theta(\kappa \ln \kappa)$ based on the comparision between $N$-$Q$-loop in Corollary 3 and $N$-loop in Corollary 1.

**In deep learning.** Also note that in the setting where the matrix-vector complexity dominates the gradient complexity, e.g., in deep learning, such two choices of $Q$ do not affect the total computational complexity. However, a smaller $Q$ can help reduce the per-iteration load on the computational resource and memory, and hence is preferred in practical applications with large models.

**Comparison among four implementations.** By comparing the complexity results in Corollaries 1, 2, 3 and 4, it can be seen that $N$-$Q$-loop and $N$-loop (both with a large $N = \Theta(\kappa \ln \kappa)$) achieve the best matrix-vector complexity $\widetilde{\mathcal{O}}(\kappa^4 \epsilon^{-1})$, whereas $Q$-loop and No-loop (both with a smaller $N = 1$) require higher matrix-vector complexity of $\widetilde{\mathcal{O}}(\kappa^6 \epsilon^{-1})$. Also note that $N$-$Q$-loop has the lowest gradient complexity. This suggests that the introduction of the inner loop with large $N$ can help to reduce the total computational complexity.

## 5 Convergence Analysis of ITD-BiO

In this section, we first provide a unified theory for ITD-BiO, which is applicable for all choices of $N$, and then specialize the convergence theory to characterize the computational complexity for the two implementations of ITD-BiO: No loop and $N$-$N$-loop. We also provide a convergence lower bound to justify the necessity of choosing large $N$ to achieve a vanishing convergence error. The following theorem characterizes the convergence rate of ITD-BiO for all choices of $N$.

**Theorem 3.** *Suppose Assumptions 1, 2, 3 and 4 hold. Define* $w = \Theta\left(\frac{\kappa^2}{\alpha\mu}(1-\alpha\mu)^N \lambda_N + \frac{w_N^2}{\mu^2}\right)$ *and* $\tau = (\lambda_N + N^2)(1-\alpha\mu)^N + w_N^2$, *where* $\lambda_N$ *and* $w_N$ *are given by* $\lambda_N = \Theta\left(\frac{w_N^2 + (1+\alpha L N)^2}{1 - \frac{1}{4}\alpha\mu - (1-\alpha\mu)^N(1+\frac{1}{2}\alpha\mu)}\right)$, $w_N = \Theta\left(\left(1 + \frac{\alpha(1 - (1-\alpha\mu)^{\frac{N}{2}})}{1 - \sqrt{1-\alpha\mu}}\right)\frac{\alpha(1-(1-\alpha\mu)^{\frac{N}{2}})}{1-\sqrt{1-\alpha\mu}}(1-\alpha\mu)^{\frac{N}{2}-1}\right)$. *Choose parameters such that* $\beta^2 \leq \frac{1-\frac{1}{4}\alpha\mu}{2w}, \alpha \leq \frac{1}{2L}$ *and* $\beta L_\Phi + \frac{8}{\alpha\mu}\left(\frac{1}{2} + \beta L_\Phi\right)w\beta^2 < \frac{1}{4}$, *where* $L_\Phi = \Theta(\kappa^3)$ *denotes the smoothness parameter of* $\Phi(\cdot)$. *Then, we have*

$$\frac{1}{K}\sum_{k=0}^{K-1}\|\nabla\Phi(x_k)\|^2 = \mathcal{O}\left(\frac{\Delta_\Phi}{\beta K} + \frac{\tau\Delta_y}{\mu^2 K} + \frac{(1-\alpha\mu)^{2N}}{\mu^3 K} + \frac{M^2(1-\alpha\mu)^{2N}L^2}{\alpha\mu^3}\right), \qquad (5)$$

*where* $\Delta_\Phi = \Phi(x_0) - \min_x \Phi(x)$ *and* $\Delta_y = \|y_0 - y^*(x_0)\|^2$.

The complete version of Theorem 3 with full parameter specifications can be found in Appendix N. In Theorem 3, the upper bound on the convergence rate for ITD-BiO contains a convergent term $\mathcal{O}\left(\frac{1}{K}\right)$ (which converges to zero sublinearly with $K$) and an error term $\mathcal{O}\left(\frac{M^2(1-\alpha\mu)^{2N}}{\alpha\mu^3}\right)$ (which is independent of $K$, and possibly non-vanishing if $N$ is chosen to be small). To show that such a possibly non-vanishing error term (when $N$ is chosen to be small) fundamentally exists, we next provide the following lower bound on the convergence rate of ITD-BiO.

**Theorem 4** (**Lower Bound**). *Consider the ITD-BiO algorithm in Algorithm 2 with* $\alpha \leq \frac{1}{L}$, $\beta \leq \frac{1}{L_\Phi}$ *and* $N \leq \mathcal{O}(1)$, *where* $L_\Phi$ *is the smoothness parameter of* $\Phi(x)$. *There exist objective functions* $f(x,y)$ *and* $g(x,y)$ *that satisfy Assumptions 1, 2, 3 and 4 such that for all iterates* $x_K$ *(where* $K \geq 1$) *generated by ITD-BiO in Algorithm 2,* $\|\nabla\Phi(x_K)\|^2 \geq \Theta\left(\frac{L^2 M^2}{\mu^2}(1-\alpha\mu)^{2N}\right)$.

Clearly, the error term in the upper bound given in Theorem 3 matches the lower bound given in Theorem 4 in terms of $\frac{M^2 L^2}{\mu^2}(1-\alpha\mu)^{2N}$, and there is still a gap on the order of $\alpha\mu$, which requires future efforts to address. Theorem 3 and Theorem 4 together indicate that in order to achieve an $\epsilon$-accurate stationary point, $N$ has to be chosen as large as $N = \Theta(\kappa\log\frac{\kappa}{\epsilon})$. This corresponds to the $N$-$N$-loop implementation of ITD-BiO, where large $N$ achieves a highly accurate hypergradient estimation in each step. Another No-loop implementation chooses a small constant-level $N = \Theta(1)$ to achieve an efficient execution per step, where a large $N$ can cause large memory usage and computation cost. Following from Theorem 3 and Theorem 4, such No-loop implementation necessarily suffers from a non-vanishing error.

In the following corollaries, we further specialize Theorem 3 to obtain the complexity analysis for ITD-BiO under the two aforementioned implementations of ITD-BiO.

**Corollary 5** ($N$-$N$-loop). *Consider* $N$-$N$-loop ITD-BiO with $N = \Theta(\kappa\ln\frac{\kappa}{\epsilon})$. *Under the same setting of Theorem 3, choose* $\beta = \min\left\{\sqrt{\frac{\alpha\mu}{40w}}, \sqrt{\frac{1-\frac{\alpha\mu}{4}}{2w}}, \frac{1}{8L_\Phi}\right\}$, $\alpha = \frac{1}{2L}$. *Then,* $\frac{1}{K}\sum_{k=0}^{K-1}\|\nabla\Phi(x_k)\|^2 = \mathcal{O}\left(\frac{\kappa^3}{K} + \epsilon\right)$, *and the complexity is* $\text{Gc}(\epsilon) = \widetilde{\mathcal{O}}(\kappa^4\epsilon^{-1})$, $\text{MV}(\epsilon) = \widetilde{\mathcal{O}}(\kappa^4\epsilon^{-1})$.

Corollary 5 shows that for a large $N = \Theta(\kappa\ln\frac{\kappa}{\epsilon})$, we can guarantee that ITD-BiO converges to an $\epsilon$-accurate stationary point, and the gradient and matrix-vector product complexities are given by $\widetilde{\mathcal{O}}(\kappa^4\epsilon^{-1})$. We note that [19] also analyzed the ITD-BiO with $N = \Theta(\kappa\ln\frac{\kappa}{\epsilon})$, and provided the same complexities as our results in Corollary 5. In comparison, our analysis has several differences. First, [19] assumed that the minimizer $y^*(x_k)$ at the $k^{th}$ iteration is bounded, whereas our analysis does not impose this assumption. Second, [19] involved an **additional** error term $\max_{k=1,...,K}\|y^*(x_k)\|\frac{L^2 M^2(1-\alpha\mu)^N}{\mu^4}$, which can be very large (or even unbounded) under standard Assumptions 1, 2, 3 and 4. We next characterize the convergence for the small $N = \Theta(1)$.

**Corollary 6** (No-loop). *Consider No-loop ITD-BiO with* $N = \Theta(1)$. *Under the same setting of Theorem 3, choose stepsizes* $\alpha = \frac{1}{2NL}$ *and* $\beta = \min\left\{\sqrt{\frac{\alpha\mu}{40w}}, \sqrt{\frac{1-\frac{\alpha\mu}{4}}{2w}}, \frac{1}{8L_\Phi}\right\}$. *Then, we have* $\frac{1}{K}\sum_{k=0}^{K-1}\|\nabla\Phi(x_k)\|^2 = \mathcal{O}\left(\frac{\kappa^3}{K} + \frac{M^2 L^2}{\alpha\mu^3}\right)$.

Corollary 6 indicates that for the constant-level $N = \Theta(1)$, the convergence bound contains a non-vanishing error $\mathcal{O}\left(\frac{M^2 L^2}{\alpha\mu^3}\right)$. As shown in the convergence lower bound in Theorem 4, under

standard Assumptions 1, 2, 3 and 4, such an error is unavoidable. Comparison between the above two corollaries suggests that for ITD-BiO, the $N$-$N$-loop is necessary to guarantee a vanishing convergence error, whereas No-loop necessarily suffers from a non-vanishing convergence error.

## 6 Empirical Verification

**Experiments on hyperparameter optimization on MNIST.** We first conduct experiments to verify our theoretical results in Corollaries 1, 2, 3 and 4 on AID-BiO with different implementations. We consider the following hyperparameter optimization problem.

$$\min_{\lambda} \mathcal{L}_{\mathcal{D}_{\text{val}}}(\lambda) = \frac{1}{|\mathcal{D}_{\text{val}}|} \sum_{\xi \in \mathcal{D}_{\text{val}}} \mathcal{L}(w_*; \xi), \quad \text{s.t. } w^* = \arg\min_{w} \frac{1}{|\mathcal{D}_{\text{tr}}|} \sum_{\xi \in \mathcal{D}_{\text{tr}}} \left( \mathcal{L}(w; \xi) + \frac{\lambda}{2} \|w\|_2^2 \right),$$

where $\mathcal{D}_{\text{tr}}$ and $\mathcal{D}_{\text{val}}$ stand for training and validation datasets, $\mathcal{L}(w; \xi)$ denotes the loss function induced by the model parameter $w$ and sample $\xi$, and $\lambda > 0$ denotes the regularization parameter. The goal is to find a good hyperparameter $\lambda$ to minimize the validation loss evaluated at the optimal model parameters for the regularized empirical risk minimization problem.

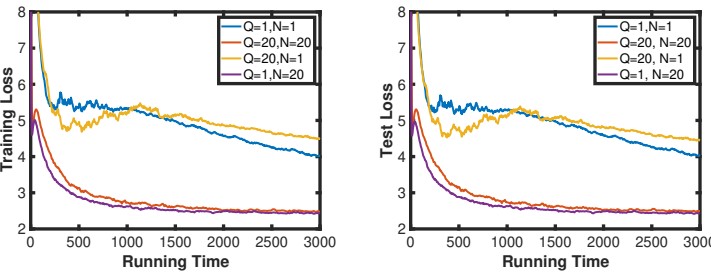

Figure 1: Training & test losses v.s. time (seconds) by AID-BiO on MNIST with different $Q$ and $N$.

From Figure 1, we can make the following observations for AID-BiO. First, the learning curves with $N = 20$ are significantly better than those with $N = 1$, indicating that running multiple steps of gradient descent in the inner loop (i.e., $N > 1$) is crucial for fast convergence. This observation is consistent with our complexity result that $N$-loop is better than No-loop, and $N$-$Q$-loop is better than $Q$-loop, as shown in Table 1. The reason is that a more accurate hypergradient estimation can accelerate the convergence rate and lead to a reduction on the Jacobian- and Hessian-vector computational complexity. Second, $N$-$Q$-loop ($N = 20$, $Q = 20$) and $N$-loop ($N = 20$, $Q = 1$) achieve a comparable convergence performance, and a similar observation can be made for $Q$-loop ($N = 1$, $Q = 20$) and No-loop ($N = 1$, $Q = 1$). This is also consistent with the complexity result provided in Table 1, where different choices of $Q$ do not affect the **dominant** matrix-vector complexity.

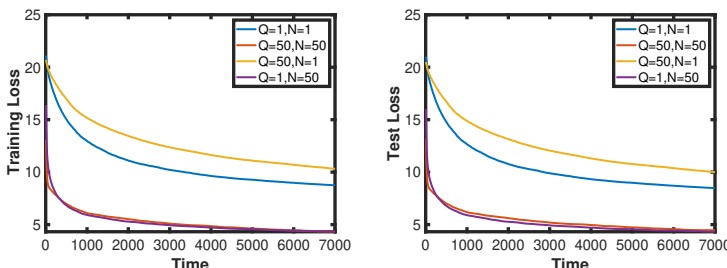

Figure 2: Training & test losses v.s. time (seconds) by AID-BiO on MNIST with different $Q$ and $N$.

In Figure 2, we plot the training and test losses versus running time for AID-BiO, where we consider a hyperparameter optimization problem on MNIST as in Figure 1 and choose loop sizes $Q$ and $N$ from $\{1, 50\}$. Similarly to Figure 1, it can be observed that the empirical results in Figure 2 are also in consistence with our theoretical results in Table 1.

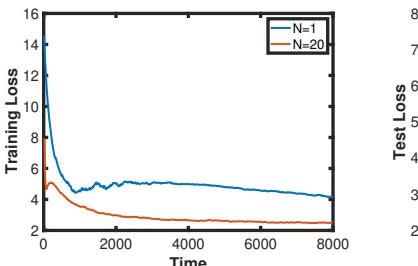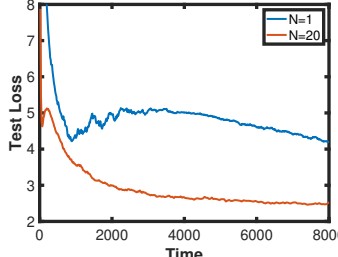

Figure 3: Training & test losses v.s. time (seconds) by ITD-BiO on MNIST with different $N$.

In Figure 3, we plot the performance of ITD-BiO with different choices of $N$ from $\{1, 20\}$ on the hyperparameter optimization on MNIST. Figure 3 illustrates that $N$-loop ITD-BiO (i.e., N=20) converges to a much smaller loss value than No-loop ITD-BiO (i.e., N=1). This is in consistence with our thereotical results in Table 2.

**Experiments on hyper-representation.** We consider a hyper-representation problem in [39], where the inner problem is to find optimal regression parameters $w$ and the outer procedure is to find the best representation parameters $\lambda$. In specific, the bilevel problem takes the following form:

$$\min_{\lambda} \Phi(\lambda) = \frac{1}{2p} \left\| h(X_V; \lambda)w^* - Y_V \right\|^2, \text{ s.t. } w^* = \operatorname*{argmin}_{w} \frac{1}{2q} \|h(X_T; \lambda)w - Y_T\|^2 + \frac{\gamma}{2} \|w\|^2$$

where $X_T \in \mathbb{R}^{q \times m}$ and $X_V \in \mathbb{R}^{p \times m}$ are synthesized training and validation data, $Y_T \in \mathbb{R}^q$, $Y_V \in \mathbb{R}^p$ are their response vectors, and $h(\cdot)$ is a linear transformation. The generation of $X_T, X_V, Y_T, Y_V$ and the experimental setup follow from [39]. For ITD-BiO, we choose $N = 20$ for $N$-$N$-loop ITD and $N = 1$ for No-loop ITD. The results are reported with the best-tuned hyperparameters.

| Algorithm | $k = 10$ | $k = 50$ | $k = 100$ | $k = 500$ | $k = 1000$ |
|---|---|---|---|---|---|
| $N$-$N$-loop ITD | 9.32 | 0.11 | 0.01 | **0.004** | **0.004** |
| No-loop ITD | 435 | 6.9 | 0.04 | 0.04 | 0.04 |

Table 3: Validation loss v.s. the number of iterations for ITD-based algorithms.

Table 3 indicates that $N$-$N$-loop with $N = 20$ can achieve a small loss value of $0.004$ after 500 total iterations, whereas No-loop with $N = 1$ converges to a much larger loss value of $0.04$. This is in consistence with our theoretical results in Table 2, where $N = 1$ can cause a non-vanishing error.

We also conduct the experiment for AID-BiO, where we choose $N$ and $Q$ from $\{1, 20\}$ for four different loop implementations. We present the results for AID-BiO in Appendix F, which also support our theoretical results in Table 1.

# 7 Conclusion

In this paper, we study two popular bilevel optimizers AID-BiO and ITD-BiO, whose implementations potentially involve additional loops of iterations within their base-loop update. By developing unified convergence analysis for all choices of the loop parameters, we are able to provide formal comparison among different implementations. Our result suggests that $N$-loops are beneficial for better computational efficiency for AID-BiO and for better convergence accuracy for ITD-BiO. This is in contrast to conventional minimax optimization, where No-loop (i.e., single-base-loop) scheme achieves better computational efficiency. Our analysis techniques can be useful to study other bilevel optimizers such as stochastic optimizers and variance reduced optimizers.

## Acknowledgements

The work of Kaiyi Ji and Lei Ying is supported in part by NSF under grants 2002608, 2001687, 2112471, 2134081, and 2207548. The work of Yingbin Liang was supported in part by the U.S. National Science Foundation under the grants CCF-1909291, DMS-2134145, and CNS-2112471.

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
