# Supplementary Materials

## A  Expanded Related Work

**Gradient-based bilevel optimization.** A number of gradient-based bilevel algorithms have been proposed via AID- and ITD-based hypergradient approximations. For example, AID-based hypergradient computation [4, 33, 10, 11, 19] estimates the Hessian-inverse-vector product by solving a linear system with an efficient iterative algorithm. ITD-based hypergradient computation [31, 8, 9, 6, 35, 17] involves a backpropagation over the inner-loop gradient-based optimization path. Convergence rate of AID- and ITD-based bilevel methods has been studied recently. For example, [10, 19] and [19, 17] analyzed the convergence rate and complexity of AID- and ITD-based bilevel algorithms, respectively. [18] characterized the lower complexity bounds for a class of gradient-based bilevel algorithms. As we mentioned before, previous studies on the convergence rate of deterministic AID-BiO [10, 19] focused only on $N$-$Q$-loop, and the only convergence rate analysis on ITD-BiO [19] was for $N$-$N$-loop. Our study here develops unified convergence analysis for all $N$ and $Q$ regimes.

Some works [30, 28, 25, 38] studied the convex inner-level objective function with multiple minimizers. [29] proposed an initialization auxiliary method for the setting where the inner-level problem is generally nonconvex.

**Stochastic bilevel optimization.** A variety of stochastic bilevel optimization algorithms have been proposed recently. For example, [10, 15, 19] proposed stochastic gradient descent (SGD) type of bilevel algorithms, and analyzed their convergence rate and complexity. Some works [13, 12, 40, 21, 3] then further improved the complexity of SGD type methods using techniques such as variance reduction, momentum acceleration and adaptive learning rate. [39] proposed a Hessian-free stochastic Evolution Strategies (ES)-based bilevel algorithm with performance guarantee. [16] proposed several algorithms for escaping saddle points in bilevel optimization. Although our study mainly focuses on determinstic bilevel optimization, our techniques can be extended to provide refined analysis for stochastic bilevel optimization to capture the order scaling with $\kappa$, which is not captured in most of the above studies on stochastic bilevel optimization.

**Bilevel optimization for machine learning.** Bilevel optimization has shown promise in many machine learning applications such as hyperparameter optimization [33, 9, 19] and few-shot meta-learning [6, 37, 34, 9, 1, 17, 20]. For example, [37, 1] introduced an outer-level procedure to learn a common embedding model for all tasks. [17] analyzed the convergence rate for meta-learning with task-specific adaptation on partial parameters.

## B  Discussion on Setting with Small Response Jacobian.

Our results in Theorem 3 and Theorem 4 apply to the general functions whose first- and second-order derivatives are Lipschitz continuous, i.e., under Assumptions 2 and 3. Here, we further discuss the extension of our results to another setting where the response Jacobian is extremely small. This setting occurs in some deep learning applications [6, 17], where the response Jacobian $\frac{\partial y^*(x)}{\partial x}$ (which is estimated by $\frac{\partial y_k^N(x)}{\partial x}$ with a large $N$) can be order-of-magnitude smaller than network gradients. Based on eq. (60) and eq. (62) in the appendix, it can be shown that the convergence error is proportional to the quantity $\frac{1}{K}\sum_{k=0}^{K-1}\|\frac{\partial y^*(x_k)}{\partial x_k}\|^2$, and hence the constant-level $N = \Theta(1)$ can still achieve a small error in this setting.

## C  Discussion on Hyperparameter Selection and Stochastic Extension.

For all loop-sizes, we set the hyperparameters to achieve the best complexity as long as the convergence is guaranteed. Let us elaborate on $N$-loop (Corollary 1) and No-loop (Corollary 2). **At a proof level**, $\lambda$ needs to satisfy $(1 - \alpha\mu)^N(1 + \lambda) < 1$ (see Lemma 2) to guarantee the convergence; otherwise the inner-loop error will explode. Given this requirement, for $N$-loop with $N = \Theta(\kappa\log\kappa)$, $\lambda = \Theta(1)$ achieves the best complexity. However, for No-loop with $N = 1$, the requirement becomes $(1 - \alpha\mu)(1 + \lambda) < 1$, and $\lambda = \Theta(\mu)$ achieves the best complexity. The stepsize $\eta$ appears in $(1 - \alpha\mu)^N\frac{\eta}{\mu}\|y_{k-1}^N - y_{k-1}^*\|^2)$ (see Lemma 1) of the error $\|v_k^Q - v_k^*\|^2$. Given the requirement $(1 - \alpha\mu)^N\frac{\eta}{\mu} < 1$, for $N$-loop with $N = \Theta(\kappa\log\kappa)$, $\eta = \Theta(1)$ achieves the best complexity, whereas

for No-loop with $N = 1$, the best $\eta = \Theta(\mu)$. **At a conceptual level**, estimating the hypergradient and linear system contains the inner-loop error $\|y_k^N - y_k^*\|^2$. For $N = 1$, the per-iteration error is large, and hence we need smaller stepsizes $\lambda, \eta, \beta$ to ensure the accumulated error not to explode. A similar argument holds for $N$-$Q$-loop and $Q$-loop.

**Extension to the stochastic setting.** If the mini-batch size at each iteration is chosen at an order of $\epsilon^{-1}$, we have checked that our proof flow and comparisons still hold.

## D  Comparison to the Analysis in [2]

We note that a similar conclusion for AID-BiO (e.g., $N = \mathcal{O}(\kappa)$ is better than $N = \mathcal{O}(1)$) has also been drawn in [2] for the stochastic bilevel optimization. The theoretical comparison in [2] focuses only on the case $Q = \Theta(\kappa)$, where each algorithm solves the linear system to a good accuracy with a large $Q$ loop. As a comparison, our theoretical comparison is more general by considering both $Q = O(\kappa)$ and $Q = O(1)$. In addition, we also provide a comparison between $Q = O(1)$ and $Q = O(\kappa)$ given different $N$, which is not covered in [2]. Also note that the choice of $Q = O(1)$ (not covered in [2]) is more challenging to analyze due to the nonvanishing error for $Q$ loop, and is more often adopted in practice, as seen in NAS [42] ($Q = 1$), meta-learning [34] ($Q = 5$), hyper-data cleaning [19] ($Q = 3$).

## E  Further Specifications on Hyperparameter Optimization Experiments

We follow the setting of [40] to setup the experiment. We first randomly sample $20000$ training samples and $10000$ test samples from MNIST dataset [24] with $10$ classes, and then add a label noise on $10\%$ of the data. The label noise is uniform across all labels from label $0$ to label $9$. We test algorithms with different values of $Q$ and $N$ to verify our theoretical results. Every algorithm's learning rates for inner and outer loops are tuned from the range of $\{0.1, 0.01, 0.001\}$ and we report the result with the best-tuned learning rates. We run $5$ random seeds and report the average result. All experiments are run over a single NVIDIA Tesla P100 GPU. The implementations of our experiments are based on the code of [19], which is under MIT License.

## F  More Experiments on Hyper-Representation

In Figure 4, we plot the outer loss of AID-BiO versus the number of matrix-vector products (i.e., MV) and the number of gradient computations (i.e., GC) on the same hyper-representation problem as in Table 3.

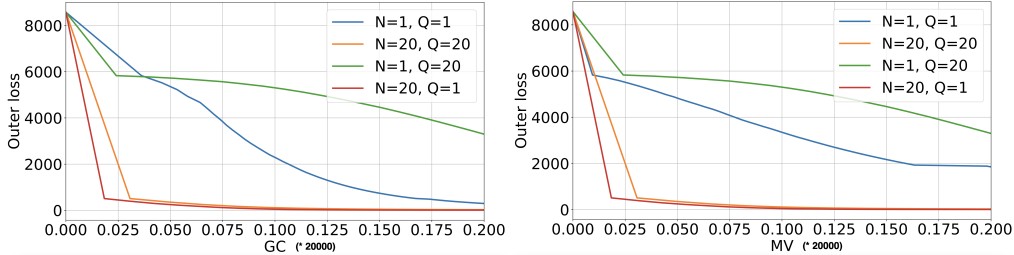

Figure 4: Outer losses of AID-BiO v.s. MV and GC on MNIST with different $Q$ and $N$.

Similarly to the observation in Figure 1, the curve with $N = 20$ significantly outperforms $N = 1$, while different choices of $Q$ do not affect the dominant matrix-vector complexity. This is consistent with our theoretical results provided in Table 1.

## G  Proof Sketch of Theorem 1

The proof of Theorem 1 contains three major steps, which include 1) decomposing the hypergradient approximation error into the $N$-loop error in estimating the inner-level solution and the $Q$-loop error

in solving the linear system approximately, 2) upper-bounding such two types of errors based on the hypergradient approximation errors at previous iterations, and 3) combining all results in the previous steps and proving the convergence guarantee. More detailed steps can be found as below.

**Step 1: decomposing hypergradient approximation error.**

We first show that the hypergradient approximation error at the $k^{th}$ iteration is bounded by

$$\|\widehat{\nabla}\Phi(x_k) - \nabla\Phi(x_k)\|^2 \leq \underbrace{\left(3L^2 + \frac{3\rho^2 M^2}{\mu^2}\right)\|y_k^* - y_k^N\|^2}_{N\text{-loop estimation error}} + \underbrace{3L^2\|v_k^* - v_k^Q\|^2}_{Q\text{-loop estimation error}}. \tag{6}$$

where the right hand side contains two types of errors induced by solving the inner-level problem and outer-level linear system. Note that for general choices of $N$ and $Q$, such two errors cannot be guaranteed to be sufficiently small, but fortunately we show via the following results that such errors contain iteratively decreasing components which facilitate the final convergence.

**Step 2: upper-bounding linear system approximation error.**

We then show that the $Q$-loop error $\|v_k^* - v_k^Q\|^2$ for solving the linear system is bounded by

$$\|v_k^Q - v_k^*\|^2 \leq \mathcal{O}\Big(\big((1+\eta\mu)(1-\eta\mu)^{2Q} + w\beta^2\big)\|v_{k-1}^Q - v_{k-1}^*\|^2$$
$$+ (\eta^2(1-\alpha\mu)^N + w\beta^2)\|y_{k-1}^* - y_{k-1}^N\|^2 + w\beta^2\|\nabla\Phi(x_{k-1})\|^2\Big). \tag{7}$$

Note that if the stepsize $\beta$ is chosen to be sufficiently small, the right hand side of eq. (7) contains an **iteratively decreasing** term $(1+\eta\mu)(1-\eta\mu)^{2Q} + w\beta^2)\|v_{k-1}^Q - v_{k-1}^*\|^2$, an error term $(\eta^2(1-\alpha\mu)^N + w\beta^2)\|y_{k-1}^* - y_{k-1}^N\|^2$ induced by the $N$-loop updates, and gradient norm term $w\beta^2\|\nabla\Phi(x_{k-1})\|^2$ that captures the increment between two adjacent iterations. Similarly, we upper-bound the $N$-loop updating error $\|y_k^* - y_k^N\|^2$ by

$$\|y_k^N - y_k^*\|^2 \leq \mathcal{O}\Big(\big((1+\lambda)(1-\alpha\mu)^N + (1+\lambda^{-1})\beta^2\big)\|y_{k-1}^N - y_{k-1}^*\|^2$$
$$+ (1+\lambda^{-1})\beta^2\|v_{k-1}^Q - v_{k-1}^*\|^2 + (1+\lambda^{-1})\beta^2\|\nabla\Phi(x_{k-1})\|^2\Big), \tag{8}$$

where $\tau = 1 + \frac{1}{\lambda}$ is inversely proportional to $\lambda$. Note that we introduce an auxiliary variable $\lambda$ in the first error term at the right hand side of eq. (8) to allow for **a general choice** of $N$. To see this, to guarantee that $(1+\lambda)(1-\alpha\mu)^N + (1+\lambda^{-1})\beta^2 < 1$, a larger $N$ allows for a smaller $\lambda$. As a result, the outer-level stepsize $\beta$ can be chosen more aggressively, which hence yields a faster convergence rate but at a cost of $N$ steps of $N$-loop updates. On the other hand, if $N$ is chosen to be small, e.g., $N = 1$, $\lambda$ needs to be as small as $\lambda = \Theta(\alpha\mu)$. As a result, $\beta$ needs to be smaller, and hence yields a slower convergence rate but with a more efficient $N$-loop update.

**Step 3: combining Steps 1 and 2.**

Combining eq. (6), eq. (7) and eq. (8), we upper-bound the hypergradient estimation error as

$$\|\widehat{\nabla}\Phi(x_k) - \nabla\Phi(x_k)\|^2 \leq \mathcal{O}\Big((1-\tau)^k + \omega\beta^2\sum_{j=0}^{k-1}(1-\tau)^j\|\nabla\Phi(x_{k-1-j})\|^2\Big),$$

which, combined with the $L_\Phi$-smoothness property of $\Phi(\cdot)$ and a proper choice of $\beta$, yields the final convergence result.

# H    Proof of Theorem 1

We first provide some auxiliary lemmas to characterize the hypergradient approximation errors.

**Lemma 1.** *Suppose Assumptions 1, 2, 3 and 4 are satisfied. Let* $v_k^* = (\nabla_y^2 g(x_k, y_k^*))^{-1} \nabla_y f(x_k, y_k^*)$ *with* $y_k^* = \arg\min_y g(x_k, y)$. *Then, we have*

$$\|v_k^Q - v_k^*\|^2 \leq (1 + \eta\mu)(1 - \eta\mu)^{2Q} \|v_{k-1}^Q - v_{k-1}^*\|^2$$

$$+ 2\left(1 + \frac{1}{\eta\mu}\right) C_Q^2 \|y_k^* - y_k^N\|^2$$

$$+ 2(1 - \eta\mu)^{2Q}\left(1 + \frac{1}{\eta\mu}\right)\left(\frac{L}{\mu} + \frac{M\rho}{\mu^2}\right)^2 \left(\frac{L}{\mu} + 1\right)^2 \|x_k - x_{k-1}\|^2,$$

*where* $C_Q = \frac{Q(1-\eta\mu)^{Q-1}\rho M\eta}{\mu} + \frac{1-(1-\eta\mu)^Q(1+\eta Q\mu)}{\mu^2}\rho M + (1 - (1-\eta\mu)^Q)\frac{L}{\mu}$.

*Proof.* Let $v_k^q$ be the $q^{th}$ $(q = 0, ..., Q-1)$ GD iterate via solving the linear system $\nabla_y^2 g(x_k, y_k^N)v = \nabla_y f(x_k, y_k^N)$, which can be written in the following iterative way.

$$v_k^{q+1} = (I - \eta\nabla_y^2 g(x_k, y_k^N))v_k^q + \eta\nabla_y f(x_k, y_k^N). \tag{9}$$

Then, by telescoping eq. (9) over $q$ from 0 to $Q$ yields

$$v_k^Q = (I - \eta\nabla_y^2 g(x_k, y_k^N))^Q v_k^0 + \eta\sum_{q=0}^{Q-1}(I - \eta_y^2 g(x_k, y_k^N))^q \nabla_y f(x_k, y_k^N). \tag{10}$$

Similarly, based on the definition of $v_k^*$, it can be derived that the following equation holds.

$$v_k^* = (I - \eta\nabla_y^2 g(x_k, y_k^*))^Q v_k^* + \eta\sum_{q=0}^{Q-1}(I - \eta_y^2 g(x_k, y_k^*))^q \nabla_y f(x_k, y_k^*). \tag{11}$$

Combining eq. (9) and eq. (10), we next characterize the difference between the estimate $v_k^Q$ and the underlying truth $v_k^*$. In specific, we have

$$\|v_k^Q - v_k^*\| \overset{(i)}{\leq} \left\|(I - \eta\nabla_y^2 g(x_k, y_k^N))^Q - (I - \eta\nabla_y^2 g(x_k, y_k^*))^Q\right\|\|v_k^*\| + (1-\eta\mu)^Q\|v_k^0 - v_k^*\|$$

$$+ \eta\left\|\sum_{q=0}^{Q-1}(I - \eta_y^2 g(x_k, y_k^N))^q - \sum_{q=0}^{Q-1}(I - \eta_y^2 g(x_k, y_k^*))^q\right\|\|\nabla_y f(x_k, y_k^*)\|$$

$$+ \eta L\left\|\sum_{q=0}^{Q-1}(I - \eta_y^2 g(x_k, y_k^N))^q\right\|\|y_k^* - y_k^N\|$$

$$\overset{(ii)}{\leq} \left\|(I - \eta\nabla_y^2 g(x_k, y_k^N))^Q - (I - \eta\nabla_y^2 g(x_k, y_k^*))^Q\right\|\frac{M}{\mu} + (1-\eta\mu)^Q\|v_{k-1}^Q - v_k^*\|$$

$$+ \eta M\left\|\sum_{q=0}^{Q-1}(I - \eta_y^2 g(x_k, y_k^N))^q - \sum_{q=0}^{Q-1}(I - \eta_y^2 g(x_k, y_k^*))^q\right\|$$

$$+ (1 - (1-\eta\mu)^Q)\frac{L}{\mu}\|y_k^* - y_k^N\|. \tag{12}$$

where $(i)$ follows from the strong convexity of $g(x, \cdot)$ and (ii) follows from Assumption 4, the warm start initialization $v_k^0 = v_{k-1}^Q$ and $\|v_k^*\| \leq \|(\nabla_y^2 g(x_k, y_k^*))^{-1}\|\|\nabla_y f(x_k, y_k^*)\| \leq \frac{M}{\mu}$. We next provide an upper bound on the quantity $\Delta_q := \|(I - \eta_y^2 g(x_k, y_k^N))^q - (I - \eta_y^2 g(x_k, y_k^*))^q\|$ in eq. (12). In specific, we have

$$\Delta_q \overset{(i)}{\leq} (1 - \eta\mu)\Delta_{q-1} + (1 - \eta\mu)^{q-1}\eta\|\nabla_y^2 g(x_k, y_k^*) - \nabla_y^2 g(x_k, y_k^N)\|$$

$$\leq (1 - \eta\mu)\Delta_{q-1} + (1 - \eta\mu)^{q-1}\eta\rho\|y_k^N - y_k^*\|. \tag{13}$$

where $(i)$ follows from the strong convexity of $g(x, \cdot)$ and Assumption 3. Telescoping eq. (13) yields

$$\Delta_q \leq (1 - \eta\mu)^q \Delta_0 + q(1 - \eta\mu)^{q-1}\eta\rho\|y_k^N - y_k^*\| = q(1 - \eta\mu)^{q-1}\eta\rho\|y_k^N - y_k^*\|,$$

which, in conjunction with eq. (12), yields

$$\|v_k^Q - v_k^*\| \leq Q(1-\eta\mu)^{Q-1}\eta\rho\frac{M}{\mu}\|y_k^N - y_k^*\| + (1-\eta\mu)^Q\|v_{k-1}^Q - v_k^*\|$$

$$+ \eta M \sum_{q=0}^{Q-1} q(1-\eta\mu)^{q-1}\eta\rho\|y_k^N - y_k^*\| + (1-(1-\eta\mu)^Q)\frac{L}{\mu}\|y_k^* - y_k^N\|. \quad (14)$$

Based on the facts that $\sum_{q=0}^{Q-1} qx^{q-1} = \frac{1-x^Q - Qx^{Q-1} + Qx^Q}{(1-x)^2} > 0$, we obtain from eq. (14) that

$$\|v_k^Q - v_k^*\| \leq \frac{Q(1-\eta\mu)^{Q-1}\rho M\eta}{\mu}\|y_k^N - y_k^*\| + (1-\eta\mu)^Q\|v_{k-1}^Q - v_{k-1}^*\|$$

$$+ (1-\eta\mu)^Q\|v_{k-1}^* - v_k^*\| + \frac{1-(1-\eta\mu)^Q(1+\eta Q\mu)}{\mu^2}\rho M\|y_k^N - y_k^*\|$$

$$+ (1-(1-\eta\mu)^Q)\frac{L}{\mu}\|y_k^* - y_k^N\|$$

which, in conjunction with $\|v_k^* - v_{k-1}^*\| \leq \left(\frac{L}{\mu} + \frac{M\rho}{\mu^2}\right)\left(\frac{L}{\mu} + 1\right)\|x_k - x_{k-1}\|$ and using the Young's inequality that $\|a+b\|^2 \leq (1+\eta\mu)\|a\|^2 + (1+\frac{1}{\eta\mu})\|b\|^2$, completes the proof of Lemma 1. $\quad\square$

**Lemma 2.** *Suppose Assumptions 1 and 2 are satisfied.*

$$\|y_k^* - y_k^N\|^2 \leq (1-\alpha\mu)^N(1+\lambda)\|y_{k-1}^N - y_{k-1}^*\|^2 + (1-\alpha\mu)^N\left(1+\frac{1}{\lambda}\right)\frac{L^2}{\mu^2}\|x_k - x_{k-1}\|^2$$

$$(15)$$

*where $\lambda$ is a positive constant.*

*Proof.* Note that $y_k^* = \arg\min_y g(x_k, y)$. Using the strong convexity (i.e., Assumption 1) and smoothness (i.e., Assumption 2) of $g(x_k, \cdot)$, we have

$$\|y_k^N - y_k^*\|^2 \leq (1-\alpha\mu)^N\|y_k^0 - y_k^*\|^2, \quad (16)$$

which, in conjunction with the warm start initialization $y_k^0 = y_{k-1}^N$ and using the Young's inequality, yields

$$\|y_k^N - y_k^*\|^2 \leq (1+\lambda)(1-\alpha\mu)^N\|y_{k-1}^N - y_{k-1}^*\|^2 + \left(1+\frac{1}{\lambda}\right)(1-\alpha\mu)^N\|y_{k-1}^* - y_k^*\|^2$$

$$\overset{(i)}{\leq} (1+\lambda)(1-\alpha\mu)^N\|y_{k-1}^N - y_{k-1}^*\|^2 + \left(1+\frac{1}{\lambda}\right)(1-\alpha\mu)^N\frac{L^2}{\mu^2}\|x_{k-1} - x_k\|^2,$$

$$(17)$$

where $(i)$ follows from Lemma 2.2 in [10]. $\quad\square$

**Lemma 3.** *Suppose Assumptions 1, 2, 3 and 4 are satisfied. Choose parameters such that $(1+\lambda)(1-\alpha\mu)^N(1+4r(1+\frac{1}{\eta\mu})L^2) \leq 1-\eta\mu$, where the notation $r = \frac{C_Q^2}{(\frac{\rho M}{\mu}+L)^2}$ with $C_Q$ given in Lemma 1. Then, we have the following inequality.*

$$\|\widehat{\nabla}\Phi(x_k) - \nabla\Phi(x_k)\|^2 \leq 3L^2(1-\eta\mu + 6wL^2\beta^2)^k\delta_0$$

$$+ 6wL^2\beta^2\sum_{j=0}^{k-1}(1-\eta\mu + 6wL^2\beta^2)^j\|\nabla\Phi(x_{k-1-j})\|^2, \quad (18)$$

*where $\delta_0 := \left(1+\frac{\rho^2 M^2}{L^2\mu^2}\right)\|y_0^N - y_0^*\|^2 + \|v_0^Q - v_0^*\|^2$ and the notation $w$ is given by*

$$w = \left(1+\frac{1}{\lambda}\right)(1-\alpha\mu)^N\left(1+\frac{\rho^2 M^2}{L^2\mu^2}\right)\frac{L^2}{\mu^2}$$

$$+ 4\left(1+\frac{1}{\eta\mu}\right)\frac{L^4}{\mu^2}\left(1+\frac{\rho^2 M^2}{L^2\mu^2}\right)\left(\frac{4(1-\eta\mu)^{2Q}}{\mu^2} + r(1-\alpha\mu)^N\left(1+\frac{1}{\lambda}\right)\right). \quad (19)$$

*Proof.* Combining Lemma 1 and Lemma 2, we have

$$
\begin{aligned}
\|v_k^Q - v_k^*\|^2 \leq & (1 + \eta\mu)(1 - \eta\mu)^{2Q}\|v_{k-1}^Q - v_{k-1}^*\|^2 \\
& + 2(1 - \alpha\mu)^N (1 + \lambda)\Big(1 + \frac{1}{\eta\mu}\Big) C_Q^2 \|y_{k-1}^N - y_k^*\|^2 \\
& + 2(1 - \alpha\mu)^N \Big(1 + \frac{1}{\lambda}\Big)\Big(1 + \frac{1}{\eta\mu}\Big) C_Q^2 \frac{L^2}{\mu^2}\|x_{k-1} - x_k\|^2 \\
& + 2(1 - \eta\mu)^{2Q}\Big(1 + \frac{1}{\eta\mu}\Big)\Big(\frac{L}{\mu} + \frac{M\rho}{\mu^2}\Big)^2 \Big(\frac{L}{\mu} + 1\Big)^2 \|x_k - x_{k-1}\|^2,
\end{aligned}
$$

which, in conjunction with $(\frac{L}{\mu} + 1)^2 \leq 4\frac{L^2}{\mu^2}$ and the notation $r = \frac{C_Q^2}{(\frac{\rho M}{\mu} + L)^2}$, yields

$$
\begin{aligned}
\|v_k^Q - v_k^*\|^2 \leq & (1 + \eta\mu)(1 - \eta\mu)^{2Q}\|v_{k-1}^Q - v_{k-1}^*\|^2 \\
& + 2\Big(1 + \frac{1}{\eta\mu}\Big)\frac{L^2}{\mu^2}\Big(L + \frac{\rho M}{\mu}\Big)^2 \Big(\frac{4(1 - \eta\mu)^{2Q}}{\mu^2} + r(1 - \alpha\mu)^N\Big(1 + \frac{1}{\lambda}\Big)\Big)\|x_k - x_{k-1}\|^2 \\
& + 2(1 + \lambda)(1 - \alpha\mu)^N\Big(1 + \frac{1}{\eta\mu}\Big)\Big(\frac{\rho M}{\mu} + L\Big)^2 r\|y_{k-1}^N - y_{k-1}^*\|^2.
\end{aligned} \tag{20}
$$

Then, combining Lemma 2 and eq. (20), we have

$$
\begin{aligned}
& \Big(1 + \frac{\rho^2 M^2}{L^2\mu^2}\Big)\|y_k^N - y_k^*\|^2 + \|v_k^Q - v_k^*\|^2 \\
\leq & (1 + \lambda)(1 - \alpha\mu)^N\Big(1 + \frac{\rho^2 M^2}{L^2\mu^2}\Big)\|y_{k-1}^N - y_{k-1}^*\|^2 \\
& + \Big(1 + \frac{1}{\lambda}\Big)(1 - \alpha\mu)^N\Big(1 + \frac{\rho^2 M^2}{L^2\mu^2}\Big)\frac{L^2}{\mu^2}\|x_{k-1} - x_k\|^2 \\
& + (1 + \eta\mu)(1 - \eta\mu)^{2Q}\|v_{k-1}^Q - v_{k-1}^*\|^2 \\
& + 4\Big(1 + \frac{1}{\eta\mu}\Big)(1 + \lambda)\Big(L^2 + \frac{\rho^2 M^2}{\mu^2}\Big)(1 - \alpha\mu)^N r\|y_{k-1}^N - y_{k-1}^*\|^2 \\
& + 4\Big(1 + \frac{1}{\eta\mu}\Big)\frac{L^4}{\mu^2}\Big(1 + \frac{\rho^2 M^2}{\mu^2 L^2}\Big)\Big(\frac{4(1 - \eta\mu)^{2Q}}{\mu^2} + r(1 - \alpha\mu)^N\Big(1 + \frac{1}{\lambda}\Big)\Big)\|x_{k-1} - x_k\|^2
\end{aligned}
$$

which, in conjunction with the definition of $w$ in eq. (19), yields

$$
\begin{aligned}
& \Big(1 + \frac{\rho^2 M^2}{L^2\mu^2}\Big)\|y_k^N - y_k^*\|^2 + \|v_k^Q - v_k^*\|^2 \\
\leq & (1 + \lambda)(1 - \alpha\mu)^N\Big(1 + \frac{\rho^2 M^2}{L^2\mu^2}\Big)\Big(1 + 4r\Big(1 + \frac{1}{\eta\mu}\Big)L^2\Big)\|y_{k-1}^N - y_{k-1}^*\|^2 \\
& + (1 + \eta\mu)(1 - \eta\mu)^{2Q}\|v_{k-1}^Q - v_{k-1}^*\|^2 + w\|x_{k-1} - x_k\|^2.
\end{aligned} \tag{21}
$$

For notational convenience, we define $\delta_k := \big(1 + \frac{\rho^2 M^2}{L^2\mu^2}\big)\|y_k^N - y_k^*\|^2 + \|v_k^Q - v_k^*\|^2$ as the per-iteration error induced by $y_k^N$ and $v_k^Q$. Then, recalling that $(1 + \lambda)(1 - \alpha\mu)^N(1 + 4r(1 + \frac{1}{\eta\mu})L^2) \leq 1 - \eta\mu$, we obtain from eq. (21) that

$$
\delta_k \leq (1 - \eta\mu)\delta_{k-1} + 2w\beta^2\|\nabla\Phi(x_{k-1}) - \widehat{\nabla}\Phi(x_{k-1})\|^2 + 2w\beta^2\|\nabla\Phi(x_{k-1})\|^2. \tag{22}
$$

Based on the form of $\widehat{\nabla}\Phi(x_k)$ and $\nabla\Phi(x_k)$ in eq. (3) and eq. (2), we have

$$
\begin{aligned}
\|\widehat{\nabla}\Phi(x_k) - \nabla\Phi(x_k)\|^2 \leq & 3\|\nabla_x f(x_k, y_k^*) - \nabla_x f(x_k, y_k^N)\|^2 + 3\|\nabla_x\nabla_y g(x_k, y_k^N)\|^2\|v_k^* - v_k^Q\|^2 \\
& + 3\|\nabla_x\nabla_y g(x_k, y_k^*) - \nabla_x\nabla_y g(x_k, y_k^N)\|^2\|v_k^*\|^2,
\end{aligned}
$$

which, in conjunction with Assumptions 1, 2, 3 and 4, yields

$$
\|\widehat{\nabla}\Phi(x_k) - \nabla\Phi(x_k)\|^2 \leq \Big(3L^2 + \frac{3\rho^2 M^2}{\mu^2}\Big)\|y_k^* - y_k^N\|^2 + 3L^2\|v_k^* - v_k^Q\|^2. \tag{23}
$$

Substituting eq. (23) into eq. (22) yields

$$
\delta_k \leq (1 - \eta\mu + 6wL^2\beta^2)\delta_{k-1} + 2w\beta^2\|\nabla\Phi(x_{k-1})\|^2,
$$

which, by telescoping and using eq. (23), finishes the proof. $\qquad\square$

**Proof of Theorem 1**

**Theorem 5** (Restatement of Theorem 1 with full parameter specifications)**.** *Suppose Assumptions 1, 2, 3 and 4 hold. Choose parameters $\alpha, \eta$ and $\lambda$ such that $(1 + \lambda)(1 - \alpha\mu)^N(1 + 4r(1 + \frac{1}{\eta\mu})L^2) \leq 1 - \eta\mu$, where $r = \frac{C_Q^2}{(\frac{\rho M}{\mu} + L)^2}$ with $C_Q = \frac{Q(1-\eta\mu)^{Q-1}\rho M\eta}{\mu} + \frac{1-(1-\eta\mu)^Q(1+\eta Q\mu)}{\mu^2}\rho M + (1 - (1 - \eta\mu)^Q)\frac{L}{\mu}$. Let $L_\Phi = L + \frac{2L^2 + \rho M^2}{\mu} + \frac{2\rho LM + L^3}{\mu^2} + \frac{\rho L^2 M}{\mu^3}$ be the smoothness parameter of $\Phi(\cdot)$. Let $\widetilde{w} := \frac{(1-\eta\mu)\eta\mu}{3\lambda r L^2}\left(1 + \frac{\rho^2 M^2}{L^2\mu^2}\right)\frac{L^2}{\mu^2} + \left(1 + \frac{1}{\eta\mu}\right)\left(L^2 + \frac{\rho^2 M^2}{\mu^2}\right)\left(\frac{16(1-\eta\mu)^{2Q}}{\mu^2} + \frac{4(1-\eta\mu)\eta\mu}{3\lambda L^2}\right)\frac{L^2}{\mu^2}$. Choose the outer stepsize $\beta$ such that $\beta = \min\left\{\frac{1}{12L_\Phi}, \sqrt{\frac{\eta\mu}{18L^2\widetilde{w}}}\right\}$. Then,*

$$\frac{1}{K}\sum_{k=0}^{K-1}\|\nabla\Phi(x_k)\|^2 \leq \frac{8(\Phi(x_0) - \Phi(x^*))}{\beta K} + \frac{21L^2((1 + \frac{\rho^2 M^2}{L^2\mu^2})\|y_0^*\|^2 + (\frac{3M}{\mu} + \frac{2L}{\mu}\|y_0^*\|)^2)}{\eta\mu K}.$$

*Proof.* First, based on Lemma 2 in [19], we have $\nabla\Phi(\cdot)$ is $L_\Phi$-Lipschitz, where $L_\Phi = L + \frac{2L^2 + \rho M^2}{\mu} + \frac{2\rho LM + L^3}{\mu^2} + \frac{\rho L^2 M}{\mu^3} = \Theta(\kappa^3)$. Then, we have

$$\Phi(x_{k+1}) \leq \Phi(x_k) + \langle\nabla\Phi(x_k), x_{k+1} - x_k\rangle + \frac{L_\Phi}{2}\|x_{k+1} - x_k\|^2$$

$$\leq \Phi(x_k) - \left(\frac{\beta}{2} - \beta^2 L_\Phi\right)\|\nabla\Phi(x_k)\|^2 + \left(\frac{\beta}{2} + \beta^2 L_\Phi\right)\|\nabla\Phi(x_k) - \widehat{\nabla}\Phi(x_k)\|^2$$

$$\overset{(i)}{\leq} \Phi(x_k) - \left(\frac{\beta}{2} - \beta^2 L_\Phi\right)\|\nabla\Phi(x_k)\|^2 + \left(\frac{\beta}{2} + \beta^2 L_\Phi\right)3L^2\delta_0(1 - \eta\mu + 6wL^2\beta^2)^k$$

$$+ 6wL^2\beta^2\left(\frac{\beta}{2} + \beta^2 L_\Phi\right)\sum_{j=0}^{k-1}(1 - \eta\mu + 6wL^2\beta^2)^j\|\nabla\Phi(x_{k-1-j})\|^2, \tag{24}$$

where $(i)$ follows from Lemma 3, $\delta_0$ is defined in Lemma 3 and $w$ is given by eq. (19). Then, telescoping eq. (24) over $k$ from 0 to $K - 1$, denoting $x^* = \arg\min_x \Phi(x)$ and using, we have

$$\left(\frac{\beta}{2} - \beta^2 L_\Phi\right)\sum_{k=0}^{K-1}\|\nabla\Phi(x_k)\|^2$$

$$\leq \Phi(x_0) - \Phi(x^*) + \frac{3L^2\delta_0(\frac{\beta}{2} + \beta^2 L_\Phi)}{\eta\mu - 6wL^2\beta^2}$$

$$+ 6wL^2\beta^2\left(\frac{\beta}{2} + \beta^2 L_\Phi\right)\sum_{k=0}^{K-1}\sum_{j=0}^{k-1}(1 - \eta\mu + 6wL^2\beta^2)^j\|\nabla\Phi(x_{k-1-j})\|^2$$

$$\overset{(i)}{\leq} \Phi(x_0) - \Phi(x^*) + \frac{3L^2\delta_0(\frac{\beta}{2} + \beta^2 L_\Phi)}{\eta\mu - 6wL^2\beta^2} + 6wL^2\beta^2\left(\frac{\beta}{2} + \beta^2 L_\Phi\right)\frac{\sum_{j=0}^{K-1}\|\nabla\Phi(x_j)\|^2}{\eta\mu - 6wL^2\beta^2} \tag{25}$$

where $(i)$ follows because $\sum_{k=0}^{K-1}\sum_{j=0}^{k-1}a_j b_{k-1-j} \leq \sum_{k=0}^{K-1}a_k\sum_{j=0}^{K-1}b_j$. Rearranging eq. (25) yields

$$\left(\frac{1}{2} - \beta L_\Phi - \frac{6wL^2\beta^2(\frac{1}{2} + \beta L_\Phi)}{\eta\mu - 6wL^2\beta^2}\right)\frac{1}{K}\sum_{k=0}^{K-1}\|\nabla\Phi(x_k)\|^2$$

$$\leq \frac{\Phi(x_0) - \Phi(x^*)}{\beta K} + \frac{3L^2\delta_0(\frac{1}{2} + \beta L_\Phi)}{\eta\mu - 6wL^2\beta^2}\frac{1}{K}. \tag{26}$$

Note that $(1 + \lambda)(1 - \alpha\mu)^N(1 + 4r(1 + \frac{1}{\eta\mu})L^2) \leq 1 - \eta\mu$ and $r > 1$, we have

$$3\eta^2(1 - \alpha\mu)^N\left(1 + \frac{1}{\lambda}\right) \leq \frac{1 - \eta\mu}{1 + \lambda}\frac{3\eta^2(1 + \frac{1}{\lambda})}{1 + 4r(1 + \frac{1}{\eta\mu})L^2} \leq \frac{1 - \eta\mu}{\lambda}\frac{\eta^3\mu}{rL^2}, \tag{27}$$

which, combined with the definitions of $w$ and $\widetilde{w}$ given by eq. (19) and theorem 1, yields $w \leq \widetilde{w}$. Then, since we set $6\widetilde{w}L^2\beta^2 \leq \frac{\eta\mu}{3}$ in Theorem 1, we have $\frac{6wL^2\beta^2}{\eta\mu - 6wL^2\beta^2} < \frac{6\widetilde{w}L^2\beta^2}{\eta\mu - 6\widetilde{w}L^2\beta^2} < \frac{1}{2}$, which,

combined with eq. (26), yields

$$\left(\frac{1}{4} - \frac{3}{2}\beta L_\Phi\right)\frac{1}{K}\sum_{k=0}^{K-1}\|\nabla\Phi(x_k)\|^2 \le \frac{\Phi(x_0) - \Phi(x^*)}{\beta K} + \frac{9L^2\delta_0(\frac{1}{2} + \beta L_\Phi)}{2\eta\mu K},$$

which, in conjunction with $\beta \le \frac{1}{12L_\Phi}$, yields

$$\frac{1}{K}\sum_{k=0}^{K-1}\|\nabla\Phi(x_k)\|^2 \le \frac{8(\Phi(x_0) - \Phi(x^*))}{\beta K} + \frac{21L^2\delta_0}{\eta\mu K}. \tag{28}$$

Based on the updates of $y$ and $v$, we have

$$\|y_0^N - y_0^*\|^2 \le \|y_0^0 - y_0^*\|^2 = \|y_0^*\|^2$$

$$\|v_0^Q - v_0^*\| \le \|v_0^*\| + \|v_0^Q - (\nabla_y^2 g(x_0, y_0^N))^{-1}\nabla_y f(x_0, y_0^N)\| + \|(\nabla_y^2 g(x_0, y_0^N))^{-1}\nabla_y f(x_0, y_0^N)\|$$

$$\overset{(i)}{\le} \frac{M}{\mu} + \frac{2}{\mu}(L\|y_0^*\| + M), \tag{29}$$

where $(i)$ follows because the initialization $v_0^0 = 0$ and $y_0^0 = 0$. Substituting eq. (29) into $\delta_0 := \left(1 + \frac{\rho^2 M^2}{L^2\mu^2}\right)\|y_0^N - y_0^*\|^2 + \|v_0^Q - v_0^*\|^2$ and eq. (28), we complete the proof. $\qquad\square$

# I   Proof of Corollary 1

In this case, first note that all choices of $\eta, \alpha, \lambda$ and $N$ satisfy the conditions in Theorem 1. First recall that $r = \frac{C_Q^2}{(\frac{\rho M}{\mu} + L)^2}$, where

$$C_Q = \frac{Q(1 - \eta\mu)^{Q-1}\rho M\eta}{\mu} + \frac{1 - (1 - \eta\mu)^Q(1 + \eta Q\mu)}{\mu^2}\rho M + (1 - (1 - \eta\mu)^Q)\frac{L}{\mu},$$

which, combined with $Q = \Theta(1)$ and $\eta = \Theta(1)$, yields $C_Q^2 = \Theta(\kappa^2)$ and hence $r = \Theta(1)$. Note that $\widetilde{w} := \frac{(1 - \eta\mu)\eta\mu}{3\lambda r L^2}\left(1 + \frac{\rho^2 M^2}{L^2\mu^2}\right)\frac{L^2}{\mu^2} + \left(1 + \frac{1}{\eta\mu}\right)\left(L^2 + \frac{\rho^2 M^2}{\mu^2}\right)\left(\frac{16(1 - \eta\mu)^{2Q}}{\mu^2} + \frac{4(1 - \eta\mu)\eta\mu}{3\lambda L^2}\right)\frac{L^2}{\mu^2}$, which, combined with $\eta = \frac{1}{L}$ and $\lambda = 1$, yields $\widetilde{w} = \Theta(\kappa^3 + \kappa^7) = \Theta(\kappa^7)$. Based on the choice of $\beta$, we have

$$\beta = \min\left\{\frac{1}{12L_\Phi}, \sqrt{\frac{\eta\mu}{18L^2\widetilde{w}}}\right\} = \Theta(\kappa^{-4}).$$

Then, we have the following convergence result.

$$\frac{1}{K}\sum_{k=0}^{K-1}\|\nabla\Phi(x_k)\|^2 = \mathcal{O}\left(\frac{\kappa^4}{K} + \frac{\kappa^3}{K}\right).$$

Then, to achieve an $\epsilon$-accurate stationary point, we have $K = \mathcal{O}(\kappa^4\epsilon^{-1})$, and hence we have the following complexity results.

- Gradient complexity: $\mathrm{Gc}(\epsilon) = K(N + 2) = \widetilde{\mathcal{O}}(\kappa^5\epsilon^{-1})$.
- Matrix-vector product complexities (dominant computational cost):

$$\mathrm{MV}(\epsilon) = K + KQ = \widetilde{\mathcal{O}}\left(\kappa^4\epsilon^{-1}\right).$$

Then, the proof is complete.

# J   Proof of Corollary 2

Based on the choices of $\alpha, \lambda$ and $\eta \le \frac{1}{\mu Q}$, recalling $r = \frac{C_Q^2}{(\frac{\rho M}{\mu} + L)^2}$ and using the inequality that $(1 - x)^Q \ge 1 - Qx$ for any $0 < x < 1$, we have

$$r \le \frac{(\frac{\rho M\eta Q}{\mu} + \eta^2 Q^2\rho M + \eta QL)^2}{(\frac{\rho M}{\mu} + L)^2} \le 4\eta^2 Q^2,$$

which, in conjunction with $\eta \leq \frac{1}{128}\frac{\alpha\mu^2}{Q^2L^2}$, yields

$$(1+\lambda)(1-\alpha\mu)^N(1+4r(1+\frac{1}{\eta\mu})L^2) \leq (1+\lambda)(1-\alpha\mu)^N(1+16(1+\frac{1}{\eta\mu})\eta^2Q^2L^2)$$

$$\leq 1 - \frac{\alpha\mu}{4} \leq 1 - \eta\mu,$$

and hence all requirements in Theorem 1 are satisfied. Also, similarly to the proof of Corollary 1, we have $r = \Theta(1)$, which, combined with $\eta = \Theta(\kappa^{-2})$, yields $\widetilde{w} = \Theta(\kappa^6 + \kappa^9) = \Theta(\kappa^9)$, and hence

$$\beta = \min\left\{\frac{1}{12L_\Phi}, \sqrt{\frac{\eta\mu}{18L^2\widetilde{w}}}\right\} = \Theta(\kappa^{-6}).$$

Then, we have the following convergence result.

$$\frac{1}{K}\sum_{k=0}^{K-1}\|\nabla\Phi(x_k)\|^2 = \mathcal{O}\left(\frac{\kappa^6}{K} + \frac{\kappa^5}{K}\right).$$

Then, to achieve an $\epsilon$-accurate stationary point, we have $K = \mathcal{O}(\kappa^6\epsilon^{-1})$, and hence we have the following complexity results.

- Gradient complexity: $\text{Gc}(\epsilon) = 3K = \widetilde{\mathcal{O}}(\kappa^6\epsilon^{-1})$.

- Matrix-vector product complexities (dominant computational cost):

$$\text{MV}(\epsilon) = K + KQ = \widetilde{\mathcal{O}}\left(\kappa^6\epsilon^{-1}\right).$$

Then, the proof is complete.

# K    Proof of Theorem 2

**Theorem 6** (Restatement of Theorem 2 with full parameter specifications). *Suppose Assumptions 1, 2, 3 and 4 hold. Define $\tau = (1-\alpha\mu)^N(1+\lambda+6(1+\lambda^{-1})(L^2+\rho^2M^2\mu^{-2}+2L^2C_Q^2)L^2\beta^2\mu^{-2})$, $w = 6(1-\alpha\mu)^N(L^2+\rho^2M^2\mu^{-2}+2L^2C_Q^2)(1+\lambda^{-1})L^2\mu^{-2}$, where $C_Q$ is a positive constant defined as in Theorem 1. Choose parameters $\alpha, \beta$ such that $\tau < 1$ and $\beta L_\Phi + w\beta^2\left(\frac{1}{2} + \beta L_\Phi\right)\frac{1}{1-\tau} \leq \frac{1}{4}$ hold. Then, the output of AID-BiO satisfies*

$$\frac{1}{K}\sum_{k=0}^{K-1}\|\nabla\Phi(x_k)\|^2 \leq \frac{4(\Phi(x_0)-\Phi(x^*))}{\beta K} + \frac{3}{K}\frac{\delta_0}{1-\tau} + \frac{27L^2M^2}{\mu^2}(1-\eta\mu)^{2Q},$$

*where $\delta_0 = 3\left(L^2 + \frac{\rho^2M^2}{\mu^2} + 2L^2C_Q^2\right)(1-\alpha\mu)^N\|y_0^* - y_0\|^2$ is the initial distance.*

*Proof.* Using an approach similar to eq. (14) in Lemma 1, we have

$$\|v_k^Q - v_k^*\|^2 \leq 2C_Q^2\|y_k^* - y_k^N\|^2 + 2(1-\eta\mu)^{2Q}\|v_k^0 - v_k^*\|^2, \tag{30}$$

where $C_Q$ is defined in Lemma 1. Using the zero initialization $v_k^0$ and based on the fact that $\|v_k^*\| \leq \frac{M}{\mu}$, we obtain from eq. (30) that

$$\|v_k^Q - v_k^*\|^2 \leq 2C_Q^2\|y_k^* - y_k^N\|^2 + \frac{2(1-\eta\mu)^{2Q}M^2}{\mu^2},$$

which, in conjunction with eq. (23), yields

$$\|\widehat{\nabla}\Phi(x_k) - \nabla\Phi(x_k)\|^2 \leq \left(3L^2 + \frac{3\rho^2M^2}{\mu^2} + 6L^2C_Q^2\right)\|y_k^N - y_k^*\|^2 + \frac{6L^2(1-\eta\mu)^{2Q}M^2}{\mu^2}. \tag{31}$$

Then, substituting eq. (31) into Lemma 2, and using the definition of $\tau$ in Theorem 2, we have

$$
\begin{aligned}
\|y_k^* - y_k^N\|^2 \leq & (1-\alpha\mu)^N(1+\lambda)\|y_{k-1}^N - y_{k-1}^*\|^2 + 2(1-\alpha\mu)^N\Big(1+\frac{1}{\lambda}\Big)\frac{L^2}{\mu^2}\beta^2\|\nabla\Phi(x_{k-1})\|^2 \\
& + 2(1-\alpha\mu)^N\Big(1+\frac{1}{\lambda}\Big)\frac{L^2}{\mu^2}\beta^2\|\widehat{\nabla}\Phi(x_{k-1}) - \nabla\Phi(x_{k-1})\|^2 \\
\leq & \tau\|y_{k-1}^N - y_{k-1}^*\|^2 + 2(1-\alpha\mu)^N\Big(1+\frac{1}{\lambda}\Big)\frac{L^2}{\mu^2}\beta^2\|\nabla\Phi(x_{k-1})\|^2 \\
& + 12(1-\alpha\mu)^N\Big(1+\frac{1}{\lambda}\Big)\frac{L^4 M^2}{\mu^4}\beta^2(1-\eta\mu)^{2Q}.
\end{aligned}
\tag{32}
$$

Telescoping eq. (32) over $k$ yields

$$
\begin{aligned}
\|y_k^* - y_k^N\|^2 \leq & \tau^k\|y_0^* - y_0^N\|^2 + 2(1-\alpha\mu)^N\Big(1+\frac{1}{\lambda}\Big)\frac{L^2}{\mu^2}\beta^2\sum_{j=0}^{k-1}\tau^j\|\nabla\Phi(x_{k-1-j})\|^2 \\
& + \frac{12}{1-\tau}(1-\alpha\mu)^N\Big(1+\frac{1}{\lambda}\Big)\frac{L^4 M^2}{\mu^4}\beta^2(1-\eta\mu)^{2Q},
\end{aligned}
$$

which, in conjunction with eq. (31), $\|y_0^* - y_0^N\|^2 \leq (1-\alpha\mu)^N\|y_0 - y_0^*\|^2$, the notation of $w$ in Theorem 2 and $\delta_0 = 3\big(L^2 + \frac{\rho^2 M^2}{\mu^2} + 2L^2 C_Q^2\big)(1-\alpha\mu)^N\|y_0^* - y_0\|^2$, yields

$$
\begin{aligned}
\|\widehat{\nabla}\Phi(x_k) - \nabla\Phi(x_k)\|^2 \leq & \delta_0\tau^k + 6L^2(1-\eta\mu)^{2Q}\frac{M^2}{\mu^2} + w\beta^2\sum_{j=0}^{k-1}\tau^j\|\nabla\Phi(x_{k-1-j})\|^2 \\
& + \frac{6wL^2 M^2}{(1-\tau)\mu^2}(1-\eta\mu)^{2Q}\beta^2.
\end{aligned}
\tag{33}
$$

Then, using an approach similar to eq. (24), we have

$$
\begin{aligned}
\Phi(x_{k+1}) \leq & \Phi(x_k) - \Big(\frac{\beta}{2} - \beta^2 L_\Phi\Big)\|\nabla\Phi(x_k)\|^2 + \Big(\frac{\beta}{2} + \beta^2 L_\Phi\Big)\|\nabla\Phi(x_k) - \widehat{\nabla}\Phi(x_k)\|^2 \\
\overset{(i)}{\leq} & \Phi(x_k) - \Big(\frac{\beta}{2} - \beta^2 L_\Phi\Big)\|\nabla\Phi(x_k)\|^2 + \Big(\frac{\beta}{2} + \beta^2 L_\Phi\Big)\delta_0\tau^k \\
& + w\beta^2\Big(\frac{\beta}{2} + \beta^2 L_\Phi\Big)\sum_{j=0}^{k-1}\tau^j\|\nabla\Phi(x_{k-1-j})\|^2 + \frac{6L^2 M^2}{\mu^2}\Big(\frac{\beta}{2} + \beta^2 L_\Phi\Big)(1-\eta\mu)^{2Q} \\
& + \Big(\frac{\beta}{2} + \beta^2 L_\Phi\Big)\frac{6wL^2 M^2}{(1-\tau)\mu^2}(1-\eta\mu)^{2Q}\beta^2,
\end{aligned}
\tag{34}
$$

where $(i)$ follows from eq. (33). Then, rearranging the above eq. (34), we have

$$
\begin{aligned}
& \frac{1}{K}\Big(\frac{1}{2} - \beta L_\Phi\Big)\sum_{k=0}^{K-1}\|\nabla\Phi(x_k)\|^2 \\
& \leq \frac{\Phi(x_0) - \Phi(x^*)}{\beta K} + \frac{1}{K}\Big(\frac{1}{2} + \beta L_\Phi\Big)\frac{\delta_0}{1-\tau} \\
& \quad + w\beta^2\Big(\frac{1}{2} + \beta L_\Phi\Big)\frac{1}{K}\sum_{k=0}^{K-1}\sum_{j=0}^{k-1}\tau^j\|\nabla\Phi(x_{k-1-j})\|^2 + \frac{6L^2 M^2}{\mu^2}\Big(\frac{1}{2} + \beta L_\Phi\Big)(1-\eta\mu)^{2Q} \\
& \quad + \Big(\frac{1}{2} + \beta L_\Phi\Big)\frac{6wL^2 M^2}{(1-\tau)\mu^2}(1-\eta\mu)^{2Q}\beta^2,
\end{aligned}
$$

which, in conjunction with the inequality that $\sum_{k=0}^{K-1}\sum_{j=0}^{k-1}a_j b_{k-1-j} \le \sum_{k=0}^{K-1}a_k\sum_{j=0}^{K-1}b_j$, yields

$$\left(\frac{1}{2}-\beta L_\Phi - w\beta^2\left(\frac{1}{2}+\beta L_\Phi\right)\frac{1}{1-\tau}\right)\frac{1}{K}\sum_{k=0}^{K-1}\|\nabla\Phi(x_k)\|^2$$

$$\le \frac{\Phi(x_0)-\Phi(x^*)}{\beta K} + \frac{1}{K}\left(\frac{1}{2}+\beta L_\Phi\right)\frac{\delta_0}{1-\tau} + \frac{6L^2 M^2}{\mu^2}\left(\frac{1}{2}+\beta L_\Phi\right)(1-\eta\mu)^{2Q}$$

$$+ \left(\frac{1}{2}+\beta L_\Phi\right)\frac{6wL^2 M^2}{(1-\tau)\mu^2}(1-\eta\mu)^{2Q}\beta^2. \tag{35}$$

Using $\beta L_\Phi + w\beta^2\left(\frac{1}{2}+\beta L_\Phi\right)\frac{1}{1-\tau} \le \frac{1}{4}$ in the above eq. (35) yields

$$\frac{1}{K}\sum_{k=0}^{K-1}\|\nabla\Phi(x_k)\|^2 \le \frac{4(\Phi(x_0)-\Phi(x^*))}{\beta K} + \frac{3}{K}\frac{\delta_0}{1-\tau} + \frac{27L^2 M^2}{\mu^2}(1-\eta\mu)^{2Q},$$

which finishes the proof. $\qquad\square$

## L   Proof of Corollary 3

Note that we choose $N = c_n\kappa\ln\kappa$ and $Q = c_q\kappa\ln\frac{\kappa}{\epsilon}$. Then, for proper constants $c_n$ and $c_q$, we have $\beta L_\Phi < \frac{1}{8}$, $C_Q = \Theta(\kappa^2)$, $\tau = \Theta(1)$ and $w\beta^2\left(\frac{1}{2}+\beta L_\Phi\right)\frac{1}{1-\tau} < \frac{1}{8}$. Then, we have

$$\frac{1}{K}\sum_{k=0}^{K-1}\|\nabla\Phi(x_k)\|^2 = \mathcal{O}\left(\frac{\kappa^3}{K}+\epsilon\right).$$

To achieve an $\epsilon$-accurate stationary point, the complexity is given by

- Gradient complexity: $\mathrm{Gc}(\epsilon) = K(N+2) = \widetilde{\mathcal{O}}(\kappa^4\epsilon^{-1})$.

- Matrix-vector product complexities (dominant cost): $\mathrm{MV}(\epsilon) = K + KQ = \widetilde{\mathcal{O}}\left(\kappa^4\epsilon^{-1}\right)$.

The proof is then complete.

## M   Proof of Corollary 4

Choose $Q = c_q\kappa\ln\frac{\kappa}{\epsilon}$. Then, for a proper selection of the constant $c_q$, we have $C_Q = \Theta(\kappa^2)$. To guarantee $6\left(1+\frac{1}{\lambda}\right)\frac{L^2}{\mu^2}\left(L^2+\frac{\rho^2 M^2}{\mu^2}+2L^2 C_Q^2\right)\beta^2 \le \frac{\alpha\mu}{4}$, we choose $\beta = \Theta(\kappa^{-4})$, which implies $1-\tau = \Theta(\alpha\mu)$. In addition, we have $w = \Theta(\kappa^7)$ and hence $\delta_0/(1-\tau) = \mathcal{O}(\kappa^5)$. Then, we have

$$\frac{1}{K}\sum_{k=0}^{K-1}\|\nabla\Phi(x_k)\|^2 = \mathcal{O}\left(\frac{\kappa^5}{K}+\frac{\kappa^4}{K}+\epsilon\right).$$

Then, to achieve an $\epsilon$-accurate stationary point, the complexity is given by

- Gradient complexity: $\mathrm{Gc}(\epsilon) = K(N+2) = \widetilde{\mathcal{O}}(\kappa^5\epsilon^{-1})$.

- Matrix-vector product complexities (dominant cost): $\mathrm{MV}(\epsilon) = K + KQ = \widetilde{\mathcal{O}}\left(\kappa^6\epsilon^{-1}\right)$.

Then, the proof is complete.

## N   Proof of Theorem 3

We first provide two useful lemmas, which are then used to prove Theorem 3.

**Lemma 4.** *Suppose Assumptions 1, 2 and 3 are satisfied. Choose inner stepsize $\alpha < \frac{1}{L}$. Then, we have*

$$\left\| \frac{\partial y_k^N}{\partial x_k} - \frac{\partial y^*(x_k)}{\partial x_k} \right\| \leq (1-\alpha\mu)^N \left\| \frac{\partial y^*(x_k)}{\partial x_k} \right\| + w_N \|y_k^0 - y^*(x_k)\|,$$

*where we define*

$$w_N = \alpha \Big( \rho + \frac{\alpha\rho L(1-(1-\alpha\mu)^{\frac{N}{2}})}{1-\sqrt{1-\alpha\mu}} \Big) (1-\alpha\mu)^{\frac{N}{2}-1} \frac{1-(1-\alpha\mu)^{\frac{N}{2}}}{1-\sqrt{1-\alpha\mu}}. \tag{36}$$

*Proof.* Based on the updates of ITD-based method in Algorithm 2, we have, for $j = 1, ...., N$,

$$\frac{\partial y_k^j}{\partial x_k} = \frac{\partial y_k^{j-1}}{\partial x_k} - \alpha\nabla_x\nabla_y g(x_k, y_k^{j-1}) - \alpha\frac{\partial y_k^{j-1}}{\partial x_k}\nabla_y^2 g(x_k, y_k^{j-1}),$$

which, in conjunction with the fact that $\frac{\partial y_k^0}{\partial x_k} = 0$, yields

$$\frac{\partial y_k^N}{\partial x_k} = -\alpha \sum_{j=0}^{N-1} \nabla_x\nabla_y g(x_k, y_k^j) \prod_{i=j+1}^{N-1} (I - \alpha\nabla_y^2 g(x_k, y_k^i)). \tag{37}$$

Then, based on the optimality condition of $y^*(x)$ and using the chain rule, we have

$$\nabla_x\nabla_y g(x_k, y^*(x_k)) + \frac{\partial y^*(x_k)}{\partial x_k}\nabla_y^2 g(x_k, y^*(x_k)) = 0,$$

which further yields

$$\frac{\partial y^*(x_k)}{\partial x_k} = \frac{\partial y^*(x_k)}{\partial x_k} \prod_{j=0}^{N-1} (I - \alpha\nabla_y^2 g(x_k, y^*(x_k)))$$

$$- \alpha \sum_{j=0}^{N-1} \nabla_x\nabla_y g(x_k, y^*(x_k)) \prod_{i=j+1}^{N-1} (I - \alpha\nabla_y^2 g(x_k, y^*(x_k))). \tag{38}$$

For the case where $N = 1$, based on eq. (37) and eq. (38), we have

$$\left\| \frac{\partial y_k^N}{\partial x_k} - \frac{\partial y^*(x_k)}{\partial x_k} \right\| \leq (1-\alpha\mu) \left\| \frac{\partial y^*(x_k)}{\partial x_k} \right\| + \alpha\rho \|y_k^0 - y^*(x_k)\|. \tag{39}$$

Next, we prove the case where $N \geq 2$. By subtracting eq. (37) by eq. (38), we have

$$\left\| \frac{\partial y_k^N}{\partial x_k} - \frac{\partial y^*(x_k)}{\partial x_k} \right\| \leq (1-\alpha\mu)^N \left\| \frac{\partial y^*(x_k)}{\partial x_k} \right\|$$

$$+ \alpha \sum_{j=0}^{N-1} \underbrace{\left\| \nabla_x\nabla_y g(x_k, y_k^j) \prod_{i=j+1}^{N-1} (I - \alpha\nabla_y^2 g(x_k, y_k^i)) - \nabla_x\nabla_y g(x_k, y^*(x_k)) \prod_{i=j+1}^{N-1} (I - \alpha\nabla_y^2 g(x_k, y^*(x_k))) \right\|}_{\Delta_j},$$

$$\tag{40}$$

where we define $\Delta_j$ for notational convenience. Note that $\Delta_j$ is upper-bounded by

$$\Delta_j \leq (1-\alpha\mu)^{N-1-j}\rho \|y_k^j - y^*(x_k)\|$$

$$+ L \underbrace{\left\| \prod_{i=j+1}^{N-1} (I - \alpha\nabla_y^2 g(x_k, y_k^i)) - \prod_{i=j+1}^{N-1} (I - \alpha\nabla_y^2 g(x_k, y^*(x_k))) \right\|}_{M_{j+1}}. \tag{41}$$

For notational simplicity, we define a quantity $M_{j+1}$ in eq. (41) for the case where the product index starts from $j + 1$. Next we upper-bound $M_{j+1}$ via the following steps.

$$M_{j+1} \leq (1-\alpha\mu)M_{j+2} + (1-\alpha\mu)^{N-j-2}\alpha\rho \|y_k^{j+1} - y^*(x_k)\|$$

$$\overset{(i)}{\leq} (1-\alpha\mu)M_{j+2} + (1-\alpha\mu)^{N-j-2}\alpha\rho(1-\alpha\mu)^{\frac{j+1}{2}} \|y_k^0 - y^*(x_k)\|$$

$$\leq (1-\alpha\mu)M_{j+2} + (1-\alpha\mu)^{N-\frac{j}{2}-\frac{3}{2}}\alpha\rho \|y_k^0 - y^*(x_k)\|, \tag{42}$$

where $(i)$ follows by applying gradient descent to the strongly-convex smooth function $g(x_k, \cdot)$. Telescoping eq. (42) further yields

$$M_{j+1} \leq (1-\alpha\mu)^{N-j-2} M_{N-1} + \sum_{i=j+2}^{N-1} (1-\alpha\mu)^{i-j-2}(1-\alpha\mu)^{N-\frac{i-2}{2}-\frac{3}{2}}\alpha\rho\|y_k^0 - y^*(x_k)\|$$

$$\leq (1-\alpha\mu)^{N-j-2} M_{N-1} + \sum_{i=0}^{N-j-3} (1-\alpha\mu)^i(1-\alpha\mu)^{N-\frac{j}{2}-\frac{i}{2}-\frac{3}{2}}\alpha\rho\|y_k^0 - y^*(x_k)\|$$

$$\leq (1-\alpha\mu)^{N-j-2}\alpha\rho(1-\alpha\mu)^{\frac{N-1}{2}}\|y_k^0 - y^*(x_k)\|$$
$$+ \sum_{i=0}^{N-j-3} (1-\alpha\mu)^{N-\frac{j}{2}+\frac{i}{2}-\frac{3}{2}}\alpha\rho\|y_k^0 - y^*(x_k)\|$$

$$\leq \sum_{i=0}^{N-j-2} (1-\alpha\mu)^{N-\frac{j}{2}+\frac{i}{2}-\frac{3}{2}}\alpha\rho\|y_k^0 - y^*(x_k)\|,$$

which, in conjunction with $\sum_{i=0}^{N-j-2}(1-\alpha\mu)^{\frac{i}{2}} \leq \frac{1-(1-\alpha\mu)^{\frac{N}{2}}}{1-\sqrt{1-\alpha\mu}}$, yields

$$M_{j+1} \leq \frac{\alpha\rho(1-(1-\alpha\mu)^{\frac{N}{2}})}{1-\sqrt{1-\alpha\mu}}(1-\alpha\mu)^{N-\frac{j}{2}-\frac{3}{2}}\|y_k^0 - y^*(x_k)\|. \tag{43}$$

Then, substituting eq. (43) into eq. (41) yields

$$\Delta_j \leq (1-\alpha\mu)^{N-1-\frac{j}{2}}\rho\|y_k^0 - y^*(x_k)\|$$
$$+ \frac{\alpha\rho L(1-(1-\alpha\mu)^{\frac{N}{2}})}{1-\sqrt{1-\alpha\mu}}(1-\alpha\mu)^{N-\frac{3}{2}-\frac{j}{2}}\|y_k^0 - y^*(x_k)\|. \tag{44}$$

Summing up eq. (44) over $j$ from 0 to $N-1$ yields

$$\sum_{j=0}^{N-1} \Delta_j \leq \left(\rho + \frac{\alpha\rho L(1-(1-\alpha\mu)^{\frac{N}{2}})}{1-\sqrt{1-\alpha\mu}}\right)\|y_k^0 - y^*(x_k)\|(1-\alpha\mu)^{\frac{N}{2}-1}\frac{1-(1-\alpha\mu)^{\frac{N}{2}}}{1-\sqrt{1-\alpha\mu}}. \tag{45}$$

Then, substituting eq. (45) into eq. (40) and using the notation $w_N$ in eq. (36), we have

$$\left\|\frac{\partial y_k^N}{\partial x_k} - \frac{\partial y^*(x_k)}{\partial x_k}\right\| \leq (1-\alpha\mu)^N \left\|\frac{\partial y^*(x_k)}{\partial x_k}\right\| + w_N\|y_k^0 - y^*(x_k)\|. \tag{46}$$

Combining eq. (39) (i.e., $N=1$ case) and eq. (46) (i.e., $N \geq 2$ case) completes the proof. $\square$

**Lemma 5.** *Suppose Assumptions 1, 2, 3 and 4 hold. Define*

$$\lambda_N = \frac{4M^2 w_N^2 + 4(1-\frac{1}{4}\alpha\mu)L^2(1+\alpha LN)^2}{1-\frac{1}{4}\alpha\mu - (1-\alpha\mu)^N(1+\frac{1}{2}\alpha\mu)}$$

*and $w = \left(1+\frac{2}{\alpha\mu}\right)\frac{L^2}{\mu^2}(1-\alpha\mu)^N\lambda_N + \frac{4M^2 w_N^2 L^2}{\mu^2}$, where $w_N$ is given in eq. (36). Let $\delta_k = \|\widehat{\nabla}\Phi(x_k) - \nabla\Phi(x_k)\|^2 + \left(\lambda_N - 4L^2(1+\alpha LN)^2\right)\|y_k^N - y^*(x_k)\|^2$ denote the approximation error at the $k^{th}$ iteration. Choose stepsizes $\beta^2 \leq \frac{1-\frac{1}{4}\alpha\mu}{2w}$ and $\alpha \leq \frac{1}{2L}$. Then, we have*

$$\delta_k \leq \left(1-\frac{1}{4}\alpha\mu\right)^k \delta_0 + J_k(1-\alpha\mu)^{2N} + 2w\beta^2 \sum_{j=0}^{k-1}\left(1-\frac{1}{4}\alpha\mu\right)^{k-1-j}\|\nabla\Phi(x_j)\|^2,$$

*where $J_k = \sum_{j=0}^{k-1}\left(1-\frac{1}{4}\alpha\mu\right)^j 4M^2\left\|\frac{\partial y^*(x_{k-j})}{\partial x_{k-j}}\right\|^2$ is related to Jacobian matrix of response function.*

*Proof.* First note that using the chain rule, $\widehat{\nabla}\Phi(x_k)$ and $\nabla\Phi(x_k)$ can be written as

$$\widehat{\nabla}\Phi(x_k) = \nabla_x f(x_k, y_k^N) + \frac{\partial y_k^N}{\partial x_k}\nabla_y f(x_k, y_k^N),$$

$$\nabla\Phi(x_k) = \nabla_x f(x_k, y^*(x_k)) + \frac{\partial y^*(x_k)}{\partial x_k}\nabla_y f(x_k, y^*(x_k)). \tag{47}$$

Subtracting two equations in eq. (47), we have

$$\|\widehat{\nabla}\Phi(x_k) - \nabla\Phi(x_k)\| \leq L\|y_k^N - y^*(x_k)\|$$
$$+ \left\|\frac{\partial y_k^N}{\partial x_k}\right\| L\|y_k^N - y^*(x_k)\| + M\left\|\frac{\partial y^*(x_k)}{\partial x_k} - \frac{\partial y_k^N}{\partial x_k}\right\|, \quad (48)$$

which, in conjunction with $\|\frac{\partial y_k^N}{\partial x_k}\| = \|\alpha \sum_{j=0}^{N-1}\nabla_x\nabla_y g(x_k, y_k^j)\prod_{i=j+1}^{N-1}(I - \alpha\nabla_y^2 g(x_k, y_k^i))\| \leq \alpha L\sum_{j=0}^{N-1}(1-\alpha\mu)^{N-1-j} \leq \alpha LN$, yields

$$\|\widehat{\nabla}\Phi(x_k) - \nabla\Phi(x_k)\| \leq L\Big(1 + \alpha LN\Big)\|y_k^N - y^*(x_k)\| + M\left\|\frac{\partial y^*(x_k)}{\partial x_k} - \frac{\partial y_k^N}{\partial x_k}\right\|$$
$$\overset{(i)}{\leq} \Big(L + \alpha L^2 N\Big)\|y_k^N - y^*(x_k)\| + M\left\|\frac{\partial y^*(x_k)}{\partial x_k}\right\|(1-\alpha\mu)^N$$
$$+ Mw_N\|y_k^0 - y^*(x_k)\|, \quad (49)$$

where $(i)$ follows from Lemma 4. Using $\|y_k^0 - y^*(x_k)\| = \|y_{k-1}^N - y^*(x_k)\| \leq \|y_{k-1}^N - y^*(x_{k-1})\| + \frac{L}{\mu}\|x_k - x_{k-1}\|$ and taking the square on both sides of eq. (49), we have

$$\|\widehat{\nabla}\Phi(x_k) - \nabla\Phi(x_k)\|^2 \leq 4L^2\Big(1 + \alpha LN\Big)^2\|y_k^N - y^*(x_k)\|^2 + 4M^2\left\|\frac{\partial y^*(x_k)}{\partial x_k}\right\|^2(1-\alpha\mu)^{2N}$$
$$+ 4M^2 w_N^2\|y_{k-1}^N - y^*(x_{k-1})\|^2 + 4M^2 w_N^2\frac{L^2}{\mu^2}\|x_k - x_{k-1}\|^2. \quad (50)$$

In the meanwhile, based on Lemma 2, we have,

$$\|y_k^N - y^*(x_k)\|^2 \leq (1-\alpha\mu)^N\Big(1 + \frac{1}{2}\alpha\mu\Big)\|y_{k-1}^N - y^*(x_{k-1})\|^2$$
$$+ \Big(1 + \frac{2}{\alpha\mu}\Big)\frac{L^2}{\mu^2}(1-\alpha\mu)^N\|x_{k-1} - x_k\|^2. \quad (51)$$

Based on $\alpha \leq \frac{1}{2L}$ and the form of $\lambda_N$ in Lemma 5, we have $\lambda_N > 4L^2(1 + \alpha LN)^2 > 0$. Then, multiplying eq. (51) by $\lambda_N$ and adding eq. (50), we have

$$\|\widehat{\nabla}\Phi(x_k) - \nabla\Phi(x_k)\|^2 + \Big(\lambda_N - 4L^2\Big(1 + \alpha LN\Big)^2\Big)\|y_k^N - y^*(x_k)\|^2$$
$$\leq \Big(1 - \frac{1}{4}\alpha\mu\Big)\Big(\lambda_N - 4L^2\Big(1 + \alpha LN\Big)^2\Big)\|y_{k-1}^N - y^*(x_{k-1})\|^2 + 4M^2\left\|\frac{\partial y^*(x_k)}{\partial x_k}\right\|^2(1-\alpha\mu)^{2N}$$
$$+ \Big(\Big(1 + \frac{2}{\alpha\mu}\Big)\frac{L^2}{\mu^2}(1-\alpha\mu)^N\lambda_N + 4M^2 w_N^2\frac{L^2}{\mu^2}\Big)\|x_k - x_{k-1}\|^2, \quad (52)$$

which, in conjunction with $\|x_k - x_{k-1}\|^2 = \beta^2\|\widehat{\nabla}\Phi(x_{k-1})\|^2 \leq 2\beta^2\|\widehat{\nabla}\Phi(x_{k-1}) - \nabla\Phi(x_{k-1})\|^2 + 2\beta^2\|\nabla\Phi(x_{k-1})\|^2$ and using the notation of $w$ in Lemma 5, yields

$$\|\widehat{\nabla}\Phi(x_k) - \nabla\Phi(x_k)\|^2 + \Big(\lambda_N - 4L^2\Big(1 + \alpha LN\Big)^2\Big)\|y_k^N - y^*(x_k)\|^2$$
$$\leq \Big(1 - \frac{1}{4}\alpha\mu\Big)\Big(\lambda_N - 4L^2\Big(1 + \alpha LN\Big)^2\Big)\|y_{k-1}^N - y^*(x_{k-1})\|^2 + 4M^2\left\|\frac{\partial y^*(x_k)}{\partial x_k}\right\|^2(1-\alpha\mu)^{2N}$$
$$+ 2\beta^2 w\|\widehat{\nabla}\Phi(x_{k-1}) - \nabla\Phi(x_{k-1})\|^2 + 2\beta^2 w\|\nabla\Phi(x_{k-1})\|^2. \quad (53)$$

Using $\beta^2 \leq \frac{1 - \frac{1}{4}\alpha\mu}{2w}$ and the notation $\delta_k = \|\widehat{\nabla}\Phi(x_k) - \nabla\Phi(x_k)\|^2 + \Big(\lambda_N - 4L^2\Big(1 + \alpha LN\Big)^2\Big)\|y_k^N - y^*(x_k)\|^2$ in the above eq. (53) yields

$$\delta_k \leq 4M^2\left\|\frac{\partial y^*(x_k)}{\partial x_k}\right\|^2(1-\alpha\mu)^{2N} + \Big(1 - \frac{1}{4}\alpha\mu\Big)\delta_{k-1} + 2w\beta^2\|\nabla\Phi(x_{k-1})\|^2. \quad (54)$$

Telescoping the above eq. (54) over $k$ yields

$$\delta_k \leq \Big(1 - \frac{1}{4}\alpha\mu\Big)^k\delta_0 + \sum_{j=0}^{k-1}\Big(1 - \frac{1}{4}\alpha\mu\Big)^j 4M^2\left\|\frac{\partial y^*(x_{k-j})}{\partial x_{k-j}}\right\|^2(1-\alpha\mu)^{2N}$$
$$+ 2w\beta^2\sum_{j=0}^{k-1}\Big(1 - \frac{1}{4}\alpha\mu\Big)^{k-1-j}\|\nabla\Phi(x_j)\|^2,$$

which, in conjunction with the definition of $J_k$, finishes the proof. $\qquad\square$

**Proof of Theorem 3**

**Theorem 7** (Restatement of Theorem 3 with parameter specifications)**.** *Suppose Assumptions 1, 2, 3 and 4 hold. Define* $w = \left(1 + \frac{2}{\alpha\mu}\right)\frac{L^2}{\mu^2}(1-\alpha\mu)^N \lambda_N + \frac{4M^2 w_N^2 L^2}{\mu^2}$ *and* $\tau = N^2(1-\alpha\mu)^N + w_N^2 + \lambda_N(1-\alpha\mu)^N$, *where* $\lambda_N$ *and* $w_N$ *are given by* $\lambda_N = \frac{4M^2 w_N^2 + 4(1-\frac{1}{4}\alpha\mu)L^2(1+\alpha LN)^2}{1 - \frac{1}{4}\alpha\mu - (1-\alpha\mu)^N(1+\frac{1}{2}\alpha\mu)}$, $w_N = \alpha\left(\rho + \frac{\alpha\rho L(1-(1-\alpha\mu)^{\frac{N}{2}})}{1-\sqrt{1-\alpha\mu}}\right)(1-\alpha\mu)^{\frac{N}{2}-1}\frac{1-(1-\alpha\mu)^{\frac{N}{2}}}{1-\sqrt{1-\alpha\mu}}$. *Choose parameters such that* $\beta^2 \leq \frac{1-\frac{1}{4}\alpha\mu}{2w}, \alpha \leq \frac{1}{2L}$ *and* $\beta L_\Phi + \frac{8}{\alpha\mu}\left(\frac{1}{2} + \beta L_\Phi\right)w\beta^2 < \frac{1}{4}$, *where* $L_\Phi = L + \frac{2L^2 + \rho M^2}{\mu} + \frac{2\rho LM + L^3}{\mu^2} + \frac{\rho L^2 M}{\mu^3}$ *denotes the smoothness parameter of* $\Phi(\cdot)$*. Then, we have*

$$\frac{1}{K}\sum_{k=0}^{K-1}\|\nabla\Phi(x_k)\|^2 \leq \mathcal{O}\left(\frac{\Delta_\Phi}{\beta K} + \frac{\tau\Delta_y}{\mu^2 K} + \frac{(1-\alpha\mu)^{2N}}{\mu^3 K} + \frac{M^2(1-\alpha\mu)^{2N}L^2}{\alpha\mu^3}\right),$$

*where* $\Delta_\Phi = \Phi(x_0) - \min_x \Phi(x)$ *and* $\Delta_y = \|y_0 - y^*(x_0)\|^2$.

*Proof.* Choose the same stepsizes $\alpha$ and $\beta$ as in Lemma 5. Then, based on the smoothness of $\Phi(\cdot)$ (i.e., Lemma 2 in [19]), we have

$$
\begin{aligned}
\Phi(x_{k+1}) \leq &\Phi(x_k) - \left(\frac{\beta}{2} - \beta^2 L_\Phi\right)\|\nabla\Phi(x_k)\|^2 + \left(\frac{\beta}{2} + \beta^2 L_\Phi\right)\|\nabla\Phi(x_k) - \widehat{\nabla}\Phi(x_k)\|^2 \\
\overset{(i)}{\leq} &\Phi(x_k) - \left(\frac{\beta}{2} - \beta^2 L_\Phi\right)\|\nabla\Phi(x_k)\|^2 + \left(\frac{\beta}{2} + \beta^2 L_\Phi\right)\delta_0\left(1 - \frac{1}{4}\alpha\mu\right)^k \\
&+ 2\left(\frac{\beta}{2} + \beta^2 L_\Phi\right)w\beta^2\sum_{j=0}^{k-1}\left(1 - \frac{1}{4}\alpha\mu\right)^{k-1-j}\|\nabla\Phi(x_j)\|^2 \\
&+ \left(\frac{\beta}{2} + \beta^2 L_\Phi\right)J_k(1-\alpha\mu)^{2N}
\end{aligned}
\tag{55}
$$

where $(i)$ follows from Lemma 5 with $\delta_k \geq \|\widehat{\nabla}\Phi(x_k) - \nabla\Phi(x_k)\|^2$. Then, telescoping the above eq. (55) over $k$ from 0 to $K-1$ yields

$$
\begin{aligned}
\left(\frac{\beta}{2} - \beta^2 L_\Phi\right)\sum_{k=0}^{K-1}\|\nabla\Phi(x_k)\|^2 \leq &\; \Phi(x_0) - \Phi(x^*) + \frac{4\beta(\frac{1}{2} + \beta L_\Phi)\delta_0}{\alpha\mu} \\
&+ \sum_{k=0}^{K-1}J_k\beta\left(\frac{1}{2} + \beta L_\phi\right)(1-\alpha\mu)^{2N} \\
&+ 2\left(\frac{\beta}{2} + \beta^2 L_\Phi\right)w\beta^2\sum_{k=0}^{K-1}\sum_{j=0}^{k-1}\left(1 - \frac{1}{4}\alpha\mu\right)^{k-1-j}\|\nabla\Phi(x_j)\|^2,
\end{aligned}
\tag{56}
$$

which, combined with $\sum_{k=0}^{K-1}\sum_{j=0}^{k-1}\left(1 - \frac{1}{4}\alpha\mu\right)^{k-1-j}\|\nabla\Phi(x_j)\|^2 \leq \frac{4}{\alpha\mu}\sum_{j=0}^{K-1}\|\nabla\Phi(x_j)\|^2$, yields

$$
\begin{aligned}
\left(\frac{1}{2} - \beta L_\Phi - \frac{8}{\alpha\mu}\left(\frac{1}{2} + \beta L_\Phi\right)w\beta^2\right)&\frac{1}{K}\sum_{k=0}^{K-1}\|\nabla\Phi(x_k)\|^2 \\
\leq &\frac{\Phi(x_0) - \Phi(x^*)}{\beta K} + \frac{4(\frac{1}{2} + \beta L_\Phi)\delta_0}{\alpha\mu K} + \left(\frac{1}{2} + \beta L_\Phi\right)(1-\alpha\mu)^{2N}\frac{1}{K}\sum_{k=0}^{K-1}J_k.
\end{aligned}
\tag{57}
$$

Based on the definition of $J_k$ in Lemma 5, we have

$$
\sum_{k=0}^{K-1}J_k = \sum_{k=0}^{K-1}\sum_{j=0}^{k-1}\left(1 - \frac{1}{4}\alpha\mu\right)^j 4M^2\left\|\frac{\partial y^*(x_{k-j})}{\partial x_{k-j}}\right\|^2 \overset{(i)}{\leq} \frac{16M^2}{\alpha\mu}\sum_{k=0}^{K-1}\left\|\frac{\partial y^*(x_k)}{\partial x_k}\right\|^2,
\tag{58}
$$

where $(i)$ follows from the inequality that $\sum_{k=0}^{K-1}\sum_{j=0}^{k-1}a_j b_{k-1-j} \leq \sum_{k=0}^{K-1}a_k\sum_{j=0}^{K-1}b_j$. Choose $\beta$ such that $\beta L_\Phi + \frac{8}{\alpha\mu}\left(\frac{1}{2} + \beta L_\Phi\right)w\beta^2 < \frac{1}{4}$. In addition, based on eq. (49), recalling the definition

that $\delta_0 = \|\widehat{\nabla}\Phi(x_0) - \nabla\Phi(x_0)\|^2 + \left(\lambda_N - 4L^2\left(1 + \alpha LN\right)^2\right)\|y_0^N - y^*(x_0)\|^2$, using the fact that $\|\frac{\partial y^*(x_0)}{\partial x_0}\| \leq \frac{L}{\mu}$, we have

$$\delta_0 \leq \mathcal{O}\Big(\big(N^2(1-\alpha\mu)^N + w_N^2 + \lambda_N(1-\alpha\mu)^N\big)\|y_0 - y^*(x_0)\|^2 + \frac{L^2M^2}{\mu^2}(1-\alpha\mu)^{2N}\Big). \quad (59)$$

Recall the definition $\tau = N^2(1-\alpha\mu)^N + w_N^2 + \lambda_N(1-\alpha\mu)^N$. Then, substituting eq. (58) and eq. (59) into eq. (57) yields

$$\frac{1}{K}\sum_{k=0}^{K-1}\|\nabla\Phi(x_k)\|^2 \leq \mathcal{O}\Big(\frac{\Phi(x_0) - \Phi(x^*)}{\beta K} + \frac{\tau\|y_0 - y^*(x_0)\|^2}{\mu^2 K} + \frac{(1-\alpha\mu)^{2N}}{\mu^3 K}$$

$$+ \frac{M^2}{\alpha\mu}\left(1-\alpha\mu\right)^{2N}\frac{1}{K}\sum_{k=0}^{K-1}\Big\|\frac{\partial y^*(x_k)}{\partial x_k}\Big\|^2\Big), \quad (60)$$

which, in conjunction with $\|\frac{\partial y^*(x)}{\partial x}\| \leq \frac{L}{\mu}$, completes the proof. $\qquad\square$

## O  Proof of Corollary 5

Based on the choice of $\alpha$ and $N$ and using $\epsilon < 1$, we have $w = \Theta(\sqrt{\epsilon}\kappa^2)$

$$\tau = \frac{(\ln\frac{\kappa}{\epsilon})^2}{\kappa^2}\sqrt{\epsilon} + \sqrt{\epsilon} + \frac{\epsilon + \sqrt{\epsilon}\kappa^2(\ln\frac{\kappa}{\epsilon})^2}{\kappa^4} = \mathcal{O}(1), \quad (61)$$

which, in conjunction with $\beta = \min\left\{\sqrt{\frac{\alpha\mu}{40w}}, \sqrt{\frac{1-\frac{\alpha\mu}{4}}{2w}}, \frac{1}{8L_\Phi}\right\}$, yields $\beta = \Theta(\kappa^{-3})$. Substituting eq. (61) and $\beta = \Theta(\kappa^{-3})$ into eq. (5) yields

$$\frac{1}{K}\sum_{k=0}^{K-1}\|\nabla\Phi(x_k)\|^2 = \mathcal{O}\Big(\frac{\kappa^3}{K} + \epsilon\Big).$$

Then, to achieve an $\epsilon$-accurate stationary point, we have $K = \mathcal{O}(\kappa^3\epsilon^{-1})$, and hence we have the following complexity results.

- Gradient complexity: $\mathrm{Gc}(\epsilon) = K(N+2) = \mathcal{O}(\kappa^4\epsilon^{-1}\ln\frac{\kappa}{\epsilon})$.
- Matrix-vector product complexities (dominant computational cost):

$$\mathrm{MV}(\epsilon) = 2KN = \mathcal{O}(\kappa^4\epsilon^{-1}\ln\frac{\kappa}{\epsilon}).$$

Then, the proof is complete.

## P  Proof of Corollary 6

Based on the choice of $\alpha$ and $N$, we have

$$w_N = \Theta(\alpha(\rho + \alpha\rho LN)N) = \Theta(1),$$

$$\lambda_N = \frac{4M^2w_N^2 + 4(1 - \frac{1}{4}\alpha\mu)L^2(1+\alpha LN)^2}{1 - \frac{1}{4}\alpha\mu - (1-\alpha\mu)^N(1+\frac{1}{2}\alpha\mu)} = \Theta(\kappa),$$

and hence $w = \Theta(\kappa^4)$ and $\tau = \Theta(\kappa)$. Then, we have $\beta = \Theta(\kappa^3)$, and hence we obtain from eq. (5) that

$$\frac{1}{K}\sum_{k=0}^{K-1}\|\nabla\Phi(x_k)\|^2 = \mathcal{O}\Big(\frac{\kappa^3}{K} + \frac{M^2L^2}{\alpha\mu^3}\Big),$$

which finishes the proof.

## Q  Proof of Theorem 4

We consider the following construction of loss functions.

$$f(x, y) = \frac{1}{2} x^T Z_x x + M \mathbf{1}^T y$$

$$g(x, y) = \frac{1}{2} y^T Z_y y - L x^T y + \mathbf{1}^T y, \tag{62}$$

where $Z_x = Z_y = \begin{bmatrix} L & 0 \\ 0 & \mu \end{bmatrix}$ and $M$ is a positive constant. First note that the minimizer of inner-level function $g(x, \cdot)$ and the total gradient $\nabla \Phi(x)$ are given by

$$y^*(x) = Z_y^{-1}(Lx - \mathbf{1}),$$

$$\nabla \Phi(x) = Z_x x + LM Z_y^{-1} \mathbf{1}. \tag{63}$$

Based on the updates of ITD-based method in Algorithm 2, we have, for $t = 0, ..., N$

$$y_k^t = y_k^{t-1} - \alpha(Z_y y_k^{t-1} - Lx_k + \mathbf{1}). \tag{64}$$

Taking the derivative w.r.t. $x_k$ on the both sides of eq. (64) yields

$$\frac{\partial y_k^t}{\partial x_k} = (I - \alpha Z_y) \frac{\partial y_k^{t-1}}{\partial x_k} + \alpha L I, \tag{65}$$

Telescoping the above eq. (65) over $t$ from 1 to $N$ and using the fact that $\frac{\partial y_k^0}{\partial x_k} = 0$, yields

$$\frac{\partial y_k^N}{\partial x_k} = \alpha L \sum_{t=0}^{N-1} (I - \alpha Z_y)^t,$$

which, in conjunction with the update $x_{k+1} = x_k - \beta \frac{\partial f(x_k, y_k^N)}{\partial x_k}$, yields

$$x_{k+1} = x_k - \beta \Big( Z_x x_k + \alpha L M \sum_{t=0}^{N-1} (I - \alpha Z_y)^t \mathbf{1} \Big). \tag{66}$$

For notational convenience, let $Z_N = \alpha \sum_{t=0}^{N-1} (I - \alpha Z_y)^t$ and $x_0 = \mathbf{1}$. Telescoping eq. (66) over $k$ from 0 to $K - 1$ yields

$$x_K = (I - \beta Z_x)^K \mathbf{1} - LM \sum_{k=0}^{K-1} (I - \beta Z_x)^k \beta Z_N \mathbf{1}$$

$$= (I - \beta Z_x)^K \mathbf{1} - LM Z_x^{-1} Z_N \mathbf{1} + LM \sum_{k=K}^{\infty} (I - \beta Z_x)^k \beta Z_N \mathbf{1}$$

$$= (I - \beta Z_x)^K \mathbf{1} - LM Z_x^{-1} Z_N \mathbf{1} + LM (I - \beta Z_x)^K Z_x^{-1} Z_N \mathbf{1}. \tag{67}$$

Rearranging the above eq. (67) yields

$$\| Z_x (x_K + LM Z_x^{-1} Z_y^{-1}) \mathbf{1} \|^2$$

$$= \big\| Z_x (I - \beta Z_x)^K \mathbf{1} + LM (I - \alpha Z_y)^N Z_y^{-1} \mathbf{1} + LM (I - \beta Z_x)^K Z_N \mathbf{1} \big\|^2$$

$$\geq L^2 M^2 \| (I - \alpha Z_y)^N Z_y^{-1} \mathbf{1} \|^2 + \big\| Z_x (I - \beta Z_x)^K \mathbf{1} \big\|^2 + L^2 M^2 \big\| (I - \beta Z_x)^K Z_N \mathbf{1} \big\|^2$$

which, in conjunction with $\alpha \leq \frac{1}{L}$, yields

$$\| \nabla \Phi(x_K) \|^2 \geq L^2 M^2 \| (I - \alpha Z_y)^N Z_y^{-1} \mathbf{1} \|^2 = \Theta \Big( \frac{L^2 M^2}{\mu^2} (1 - \alpha \mu)^{2N} \Big), \tag{68}$$

which holds for all $K$.