# OpenReview forum: "Will Bilevel Optimizers Benefit from Loops"
_NeurIPS.cc/2022/Conference — NeurIPS 2022 Accept_

### Official Review · Reviewer_jDxR · 2022-07-09

**Rating:** 6
**Confidence:** 5
**Soundness:** 3 good
**Presentation:** 3 good
**Contribution:** 3 good

**Summary:**

This paper provides a unified convergence theory to capture the computational differences among different implementations in bilevel optimization algorithms such as AID-BiO and ITD-BiO, with a focus on the different choices of inner and outer loops. By comparing different implementation, the paper draws the conclusion that different from the minmax case, in the bilevel optimization, having double loops always have some provable benefit.

**Questions:**

* The lower-bound result in Theorem 4. See [Weaknesses]. It would be helpful to clarify.

* Applicability of the results to the stochastic bilevel algorithms. See [Weaknesses]. It would be helpful to clarify.

* In practice, stochastic bilevel optimization algorithms may be used more frequently. How does the conclusion in this paper compare to the conclusion in the recent work on *stochastic* bilevel optimization algorithms; for example, Theorem 1 and Proposition 2 in [Chen et al' 2021]. It would be helpful to clarify.

    Chen et al "Closing the gap: Tighter analysis of alternating stochastic gradient methods for bilevel problems." Advances in Neural Information Processing Systems, 2021.

* The performance metrics in the experiments. See [Weaknesses]. It would be helpful to clarify.

* In the AID-Bio, is it possible to provide the lower-bound similar to ITD-Bio?

* The discussion on the setting with small response Jacobian (line 304) is not clear. What conclusion do the authors want to draw from this discussion? Is this consistent with the main results?

**Limitations:**

The limitations and potential negative societal impact of their work have been properly discussed.

**Strengths And Weaknesses:**

[Strengths]
* The paper is well written and the theoretical question studied in this paper is of practical interest.
* The paper has done a good job of summarizing the theoretical results into two tables that have an explicit dependence on $\kappa$ and $\epsilon$.
* The non-convergent result (both upper and lower bounds) of No-loop ITD-Bio is interesting, which suggests a gap between the performance of AID-Bio and ITD-Bio.

[Weaknesses]
* The lower-bound result in Theorem 4 is a bit loose. Not only because it has a large gap with the upper bound, but the worst-case example used in the proof also seems not carefully crafted. Does the lower bound also depend on $K$? Is there any possibility of improving the lower bound or upper bound?

* The paper only focuses on the comparison of different *deterministic* bilevel optimization algorithms. Does the conclusion in this paper also hold in the *stochastic* setting? If not, it would be helpful by mentioning ``deterministic'' in the title or abstract.

* It is not clear to me why the metic in the AID-Bio experiments is runtime, but that in the ITD-Bio experiments is iteration? Since the theory is mostly on the comparison on the number of iterations or MV and GC, comparing those metrics in both experiments will make more sense.

---

> ### Author Response · Authors · 2022-08-02
> **Response to reviewer jDxR**
>
> Many thanks for providing the helpful review. In the revised version, we have made the changes based on the reviewer’s comments. All the changes are highlighted by the blue-colored texts.
>
> Q1: Does the lower bound also depend on $K$?
>
> A: Yes, from the proof in Appendix R, it can be seen that it has a term decaying exponentially w.r.t. $K$ (see eq. (67)). However, we do not include it in the final lower bound because the main purpose of our lower bound is to demonstrate that the nonvanishing error in our upper bound for ITD-BiO fundamentally exists and is not crafted by our bounding techniques. We will further investigate the open problem of improving the dependence on $K$ in future study.
>
> Q2: Is there any possibility of improving the lower bound or upper bound?
>
> A: Yes, for the lower bounds, it is possible to improve the dependence on $K$ and $\kappa$ via a tighter construction of nonconvex upper-level objectives. Our upper bound development treats inner and outer variables separately in the error analysis, which may be improved if we treat them as an entire one and construct a tighter error sequence different from that in Lemma 5. However, both directions require substantial efforts due to the nested structure and nonconvexity of the objective function, which we wish to leave as the future study.
>
> Q3: The paper only focuses on the comparison of different deterministic bilevel optimization algorithms. Does the conclusion in this paper also hold in the stochastic setting? If not, it would be helpful by mentioning ``deterministic'' in the title or abstract.
>
> A: Yes, if the mini-batch size at each iteration of stochastic algorithms is chosen at an order of $\epsilon^{-1}$, we have checked that our proof flow and comparisons still hold. We have clarified it in the revision.
>
> Q4: In practice, stochastic bilevel optimization algorithms may be used more frequently. How does the conclusion in this paper compare to the conclusion in the recent work on stochastic bilevel optimization algorithms; for example, Theorem 1 and Proposition 2 in [Chen et al' 2021]. It would be helpful to clarify. Chen et al "Closing the gap: Tighter analysis of alternating stochastic gradient methods for bilevel problems." Advances in Neural Information Processing Systems, 2021.
>
> A: Great point! In Chen et al' 2021, they made a similar conclusion that $N=O(\kappa)$ is better than $N=O(1)$ under the choice of $Q=O(\kappa\log\frac{1}{\epsilon})$ (i.e., solving the linear system to a good accuracy $\epsilon$). As a comparison, our theoretical comparison is more general by considering both $Q=O(\kappa)$ and $Q=O(1)$. In addition, we also provide a comparison between $Q=O(1)$ and $Q=O(\kappa)$ given different $N$, which is not covered in Chen et al' 2021. We have added this discussion in the revised pdf.
>
> Q5: It is not clear to me why the metric in the AID-Bio experiments is runtime, but that in the ITD-Bio experiments is iteration?
>
> A: Thanks! For ITD-BiO, the goal of our experiment is to show that No-loop ITD-BiO with $N=1$ induces a larger convergence error than $N$-$N$-loop ITD-BiO with $N=1$. In other words, we compare their losses after they converge, i.e., after $500$ iterations. Therefore, using the iteration as a metric serves the purpose of this comparison.
>
> Q6: Since the theory is mostly on the comparison on the number of iterations or MV and GC, comparing those metrics in both experiments will make more sense.
>
> A: Great point! We have added a new Fig. 4 in Appendix G in the revised pdf, which plots losses versus $MV$ (number of matrix-vector products) and $GC$ (number of gradients), as suggested by the reviewer. It can be seen that these empirical results are still consistent with our theoretical results. More results on such comparisons will be added.
>
> Q7: In the AID-Bio, is it possible to provide the lower-bound similar to ITD-Bio?
>
> A: The lower bound for ITD-BiO is constructed particularly to demonstrate that the convergence error of ITD-BIO with $N=O(1)$ fundamentally exists. However, since AID-BiO does not contain convergence error, our instance used for ITD-BiO may not be tight enough. In general, the lower bound construction for AID-BiO is an interesting but very challenging task, and we would like to leave it for future study.
>
> Q8: The discussion on the setting with small response Jacobian (line 304) is not clear. What conclusion do the authors want to draw from this discussion? Is this consistent with the main results?
>
> A: Thanks for pointing this out for us! We want to convey that if we assume that the response Jacobian $\frac{\partial y^*(x)}{\partial x}$ of our bilevel objective is sufficiently small at an $\epsilon$ level, our analysis can further guarantee that the final convergence error is at an $\epsilon$ level. This does not contradict our main results, because our main results do not make this extra assumption.

---

> > ### Comment · Reviewer_jDxR · 2022-08-08
> > **Thank you for rebuttal!**
> >
> > Dear authors,
> >
> > Thanks a lot for the efforts put in the rebuttal. I went through the rebuttal and find the responses to Q1, Q3, Q4, Q5, Q6, and Q8 are satisfactory but those to Q2 and Q7 are hand-waiving. In the final version, I suggest authors move Appendix B, D, and G to the main paper since they directly support the main claim of the paper.

---

> > > ### Author Response · Authors · 2022-08-08
> > > **Thanks for the further comments!**
> > >
> > > Dear Reviewer jDxR,
> > >
> > > We thank the reviewer very much for providing further feedback and suggestions! The open problems in Q2 and Q7 require substantial efforts, which we would like to explore in the future study!  As suggested by the reviewer, we will move Appendix B, D, and G to the main paper when the page space is allowed.
> > >
> > > Best,
> > > Authors

---

### Official Review · Reviewer_Kzzc · 2022-07-09

**Rating:** 8
**Confidence:** 4
**Soundness:** 3 good
**Presentation:** 4 excellent
**Contribution:** 3 good

**Summary:**

This paper provides a unified convergence rate analysis for existing two types of widely-used bilevel optimization algorithms via so-called approximate implicit differentiation (AID) and iterative differentiation (ITD), under all different choices of loop sizes. Some interesting theoretical findings are provided as well. In particular, for AID-based approaches, two major loops exist within the base loop, i.e., a loop of size N for approximating the lower-level solution and a loop of size Q for estimating the Hessian-inverse-vector (HIV) product in the hypergradient computation. In the comparison among different loop choices, they specify their theories to several typical choices of interest in practice, including N= 1 or \kappa and Q = 1 or \kappa. Based on such comparisons, they show that the lower-level loop can improve the overall computational complexity w.r.t. both the matrix-vector and gradient computations, and the loop for HIV estimation can reduce the gradient complexity. For ITD-based method, they show via upper and lower bounds that loop size needs to be large for achieving a vanishing convergence error induced by the lower-level and HIV approximations. Some empirical results are also provided and seem to validate the theories well.

**Questions:**

The paper is limited to the deterministic setting, and I am wondering if the developed analysis can be further extended to the stochastic setting with data sampling. Can the authors have some comments or provide some guidances on this extension?

In AID-based method, people sometimes use the acceleration methods for both the lower-level and HIV approximations to achieve a better complexity. It would be great to discuss how the current analysis can be extended to such scenarios, and whether the corresponding comparison still holds?

Theorems 1, 2 and 3 seems involving complicated parameters and relations. It would be good to provide some proof outlines for readers to better understand the technical idea of this paper.


**Ethics Review Area:**

["I don’t know"]

**Limitations:**

the authors adequately addressed the limitations and potential negative societal impact of their work

**Strengths And Weaknesses:**

This work is well written and the motivation is clear to me. Bilevel optimization has attracted signification attention recently from both the deep learning and optimization communities, where AID and ITD are two widely-used methods so far. Characterizing a unified convergence guarantee for both of these two types of methods is fundamentally important because different loop schemes have been used in practice, but only some of them have guarantee. This work has done a good job in providing a tight characterization and systematic study for all these cases, which is a good contribution.

The theoretical findings are interesting and provide useful insights for practical applications. By capturing different dependences on the condition number \kappa, they demonstrate the necessity of lower-level  optimization loops in both algorithms to improve the overall complexity, majorly in terms of the Hessian- and Jacobean-vector products. In particular, for ITD-based method, the lower bound is a nice and new contribution to justify the fundamental challenging for No loop scheme in achieving a vanishing error. Theories seem to be well supported by the experiments.

The technical analysis introduces some new developments, which may be of interest to the bilevel optimization community. Existing works on the convergence rate of bilevel optimization  mainly focus on the case when the HIV is solved at a good accuracy via large Q, but this paper allows small Q by showing that the coupled error is decreasing iteratively. This kind of characterization may be used in other settings with AID types of bilevel optimization.

---

> ### Author Response · Authors · 2022-08-02
> **Response to reviewer Kzzc**
>
> Many thanks for providing the helpful review. In the revised version, we have made the changes based on the reviewer’s comments. All the changes are highlighted by the blue-colored texts.
>
> Q1: The paper is limited to the deterministic setting, and I am wondering if the developed analysis can be further extended to the stochastic setting with data sampling. Can the authors have some comments or provide some guidances on this extension?
>
> A: Many thanks! Yes, if the mini-batch size at each iteration in the stochastic setting is chosen at an order of $\epsilon^{-1}$, we have checked that our proof flow and comparisons still hold. We have clarified this in the revision.
>
> Q2: In AID-based method, people sometimes use the acceleration methods for both the lower-level and HIV approximations to achieve a better complexity. It would be great to discuss how the current analysis can be extended to such scenarios, and whether the corresponding comparison still holds?
>
> A: Great point! If we use acceleration methods for the lower-level and HIV approximations, we will achieve an improved $N=O(\sqrt{\kappa})$ and $Q=O(\sqrt{\kappa})$ for $N$-$Q$-loop, an improved $N=O(\sqrt{\kappa})$ for  $N$-loop, and an improved $Q=O(\sqrt{\kappa})$ for $Q$-loop. However, it is not clear if an improvement can be obtained for No-loop. We will investigate this comparison under acceleration methods as an interesting future topic.
>
> Q3: Theorems 1, 2 and 3 seem to involve complicated parameters and relations. It would be good to provide some proof outlines for readers to better understand the technical idea of this paper.
>
> A: Good point! We have provided a proof sketch of Theorem 1 in Appendix H. For ease of interpretation, in the revised pdf, we have also provided the simplified theorems by hiding constants in the notation $\Theta(\cdot)$, and relegated the complete theorems to the appendix.

---

### Official Review · Reviewer_KmZV · 2022-07-10

**Rating:** 4
**Confidence:** 5
**Soundness:** 3 good
**Presentation:** 3 good
**Contribution:** 2 fair

**Summary:**

This paper studies the fine-grained convergence of gradient-based algorithms for bilevel optimization problems. The focus is on establishing the unified convergence analysis for both AID-BiO and ITD-BiO and on the comparison between the rate of convergence under different numbers of steps in the inner and outer loops. Under reasonable assumptions, the paper shows that AID-BiO and ITD-BiO can benefit from double loops in certain sense.

**Questions:**

See my comments in weakness.

**Limitations:**

Yes, it addresses the limitations.

**Strengths And Weaknesses:**

**Comments - Strengths**

(+) The unified convergence theory for AID-BiO is new, which captures all choices of the inner-level gradient steps N and the outer-level Hessian-inverse-vector steps Q. The lower-bound on the nonconvex-strongly convex bilevel problem seems new.

(+) The conclusion that bilevel optimization is in contrast to minimax optimization, where no-loop gradient descent ascent (GDA) with N = 1 often outperforms (N-loop) GDA is interesting.

**Comments - Weaknesses**

*Major comments*

(-) The major concern I have is that the majority of the theory in this submission is a summary of previous convergence results in [17]. Indeed, this paper and [17] focus on the same setting, the same algorithms, and the same convergence results but now with explicit dependence on the number of steps in the inner $N$ and outer loops $Q$. While this is a good point to discuss, the technical content of this paper seems to be incremental. Therefore, this paper is more like a supplementary to [17].

(-) The second major concern I have is that the theoretical comparison between the double-loop and the single-loop is in terms of the complexity upper bound. Note that the larger upper bound may not mean worse performance since the single-loop algorithm may be harder to analyze, leading to looser worst-case performance. For example, it would be interesting to compare the lower bound of the single-loop algorithm with the upper bound of the double-loop algorithms.

(-) The third major concern is on the experiments. The experiments did not closely match the theoretical analysis. Since the separation between no-loop and double-loop is on $N={\cal O}(1)$ or $N={\cal O}(\kappa)$, choosing $N, Q$ as $1, 20$ does not serve the purposes. In addition, it may be better to plot the dependence on $MV$ and $GC$ not the runtime nor the number of iterations.

*Minor comments*

(-) A similar conclusion for AID-Bio (e.g., $N={\cal O}(\kappa)$ is better than $N={\cal O}(1)$) has already been drawn in the stochastic bilevel algorithms; see ALSET (Closing the Gap: Tighter Analysis of Alternating Stochastic Gradient Methods for Bilevel Problems). The connection and difference need to be discussed.

(-) While the paper did provide a lower bound for ITD-Bio, it seems quite loose compared with the upper bound of ITD-Bio. They differ not only in the dependence on $\kappa$ but also the dependence on $K$.

(-) The main theorems (e.g., Theorems 1, 2, and 3) in the paper are presented in a very complicated way. They depend on many irrelevant constants such as $M, \rho, r$, which make the main theorems difficult to interpret.

---

> ### Author Response · Authors · 2022-08-02
> **Response to reviewer KmZV (part 3)**
>
> Q6: The main theorems (e.g., Theorems 1, 2, and 3) in the paper are presented in a very complicated way. They depend on many irrelevant constants such as $M,\rho,r$, which make the main theorems difficult to interpret.
>
> A: Thanks! Theorems 1,2,3 are general convergence results for AID-BiO and ITD-BiO with flexible hyperparameters such as loop sizes and stepsizes. Hence, we keep all constants such as $M,\rho,r$ for the completeness. For ease of interpretation, in the revised pdf, we have provided the simplified theorems by hiding constants in the notation $\Theta(\cdot)$, and relegated the complete theorems to the appendix.
>
> Finally, we thank the reviewer again for the helpful comments for our work. If our response resolves your concerns to a satisfactory level, we kindly ask the reviewer to consider raising the rating of our work. Certainly, we are more than happy to address any further questions you may have during the discussion period.

---

> ### Author Response · Authors · 2022-08-02
> **Response to reviewer KmZV (part 2)**
>
> Q2: The second major concern I have is that the theoretical comparison between the double-loop and the single-loop is in terms of the complexity upper bound. Note that the larger upper bound may not mean worse performance since the single-loop algorithm may be harder to analyze, leading to looser worst-case performance. For example, it would be interesting to compare the lower bound of the single-loop algorithm with the upper bound of the double-loop algorithms.
>
> A: Thanks! The optimization community does appreciate and benefit from the comparison of upper bounds in understanding different algorithmic designs and parameter selections, for example, as seen from the comparison among SGD, SVRG, to SPIDER/SARAH/STORM in the well established stochastic optimization literature. We do agree that the lower bound for single-loop algorithms is very interesting, but also non-trivial due to the upper-level nonconvexity and the nested structure. However, we provide some thoughts as below.
>
> Based on our analysis of upper bounds, the single-loop algorithm (i.e., the No-loop AID with $Q=N=1$) has worse complexity because its hypergradient estimation contains an error term that is a multiplication of $Q$ error (i.e., error in solving the linear system) and $N$ error (error in solving the lower-level problem). Thus, for small $N=Q=1$, the outer stepsize is smaller than that of $N$-$Q$-loop by an order of $\kappa^2$ and hence results in a worse complexity. Based on this key insight, for the worst-case construction, we can choose a quadratic lower-level problem (similar to eq. (60) for ITD-BiO lower bound), but carefully construct a nonconvex objective to guarantee that the per-iteration estimation contains this multiplication-type error. However, deriving a tight lower bound requires substantial efforts, which we wish to leave for future study.
>
> In addition, in Appendix C, we have added some discussions on the tightness of upper bounds from the proof and conceptual perspectives, which can help to understand the performance comparison and parameter selections of single- and double-loop algorithms.
>
> Q3: The third major concern is on the experiments. The experiments did not closely match the theoretical analysis. Since the separation between no-loop and double-loop is on $N =O(1)$ or $N=O(\kappa)$, choosing $N, Q$ as $1 , 20$  does not serve the purposes. In addition, it may be better to plot the dependence on $MV$ and  $GC$  not the runtime nor the number of iterations.
>
> A: Thanks! We have added new plots (see Fig. 2 in Appendix F in the revised pdf) with $N,Q$ chosen from $\{1, 50\}$, and new plots (see Fig. 4 in Appendix G in the revised pdf) of losses versus $MV$ (number of matrix-vector products) and $GC$ (number of gradients). It can be seen that these new empirical results are still consistent with our theoretical results. More comparison results will be added.
>
> Q4: A similar conclusion for AID-Bio (e.g., $N=O(\kappa)$ is better than $N =O(1)$) has already been drawn in the stochastic bilevel algorithms; see ALSET (Closing the Gap: Tighter Analysis of Alternating Stochastic Gradient Methods for Bilevel Problems). The connection and difference need to be discussed.
>
> A: Thanks for pointing this out! In ALSET, their comparison focuses only on the case $Q=\kappa\log K$, as seen from their choice of $Q$ (in their notations, they use $N$) after eq. (59) therein. In other words, they solve the linear system to a good accuracy of $\epsilon$ with a large $Q$ loop. As a comparison, our theoretical comparison is more general by considering both $Q=O(\kappa)$ and $Q=O(1)$. In addition, we also provide a comparison between $Q=O(1)$ and $Q=O(\kappa)$ given different $N$, which is not covered in the ALSET paper.  We have added this discussion in the revised pdf.
>
> Q5: While the paper did provide a lower bound for ITD-Bio, it seems quite loose compared with the upper bound of ITD-Bio. They differ not only in the dependence on $\kappa$ but also the dependence on $K$.
>
> A: The point of this lower bound is to show that the nonvanishing convergence error in our upper bound (i.e., eq.(5)) for ITD-BiO fundamentally exists and is not crafted by our bounding techniques. Our lower bound does serve such a purpose. We believe closing the gap between upper and lower bounds is a much more challenging goal, which requires the construction of tighter worst-case objective functions, which requires substantial efforts due to the upper-level nonconvexity and the nested structure. We do think this open problem is an interesting topic for future study.

---

> ### Author Response · Authors · 2022-08-02
> **Response to reviewer KmZV (part 1)**
>
> Many thanks for providing the helpful review. In the revised version, we have made the changes based on the reviewer’s comments. All the changes are highlighted by the blue-colored texts.
>
> Q1: The major concern I have is that the majority of the theory in this submission is a summary of previous convergence results in [17]. Indeed, this paper and [17] focus on the same setting, the same algorithms, and the same convergence results but now with explicit dependence on the number of steps in the inner $N$  and outer loops $Q$ . While this is a good point to discuss, the technical content of this paper seems to be incremental. Therefore, this paper is more like a supplementary to [17].
>
> A: We respectfully disagree with this comment. Our paper has significant new technical contributions beyond those in [17], as clarified below.
>
> 1. More general results: Our results hold for all parameter ranges of $Q$ and $N$ (i.e., all types of looped algorithm implementations including single-loop, double-loop, triple-loop), whereas the results in [17] have restrictive constraints on $Q,N$ (which  require large $Q$ and $N$ and hence study only the triple-loop algorithms.) Such much more general results necessitate novel developments and tighter error analysis (as we explain in the following item 2) in order to eliminate the restrictions in [17]. For example, in eq. (30) of [17], it can be seen that (using our notations) $\delta_{N,Q}=C_1\kappa^5(1-\frac{1}{\kappa})^N + C_2\kappa(1-\frac{1}{\sqrt{\kappa}})^Q<1$ ($\delta_{N,Q}$ is defined in eq. (17) therein), which clearly means that $N$ and $Q$ have to be chosen at an order of $\kappa\log\kappa$.
>
> 2. In terms of technical developments, our analysis includes new developments and is much more challenging than that in [17], in order to accommodate the entire parameter range for $Q$ and $N$, which corresponds to different loops of implementations. We further elaborate this below.
>
>     * To characterize the error of solving the linear system for AID, we devise a recursion-based analysis (see eq. (12),(13) in appendix) to bound the error between $v_k^Q$ (derived in eq. (10)) and $v_k^*$ (derived in eq. (11)). This development does not exist in the analysis [17] (see Lemma 3 therein), and provides a tighter characterization especially when $Q$ is small due to a tighter dependence on the stepsize $\eta$ (see Lemma 1). Other big differences can be found in the error analysis of $y_k^N$ (compare our Lemma 2 with eq. (23) in [17]), the construction of error sequence (compare our eq. (22) with eq. (22) in [17]), and etc.
>
>     * For  ITD, the analysis in [17] assumes the initial gap $\|y_k^0-y^*(x_k)\|$ to be bounded (see eq. (40) therein). As a comparison, our analysis eliminates this assumption by constructing a decent sequence $\delta_k$ shown in eq. (54), which does not exist in the analysis of [17]. Furthermore, the analysis in [17] leads to a much larger error $O(\kappa^4)$ (see eq. (42) therein) for the choice of $N=O(1)$ due to a less tight characterization.
>
> 3. For ITD-based algorithms, our characterization on the lower bound and the nonvanishing error is new, which is not studied in [17].

---

> ### Author Response · Authors · 2022-08-05
> **Could you please let us know your feedback?**
>
> Dear Reviewer KmZV:
>
> Since the author-reviewer discussion period has started for a few days, we will appreciate if you could check our response to your review comments soon. This way, if you have further questions and comments, we can still reply before the author-reviewer discussion period ends. If our response resolves your concerns, we kindly ask you to consider raising the rating of our work. Thank you very much for your time and efforts!
>
> Thanks,
> Authors

---

> ### Author Response · Authors · 2022-08-08
> **Your prompt response is highly appreciated**
>
> Dear Reviewer KmZV:
>
> As the author-reviewer discussion period ends soon, we will appreciate very much if your could check our response soon. In particular, our response has explained in detail about your concerns on the contribution of the paper beyond [17] (Q1) and comparison among upper bounds (Q2). We also added new experiment plots regarding your concern on the experiment (Q3). If our response resolves your concerns, we kindly ask you to consider raising the rating of our work. We are also more than happy to answer your further questions. Thank you very much for your time and efforts!

---

### Official Review · Reviewer_rZSp · 2022-07-11

**Rating:** 7
**Confidence:** 4
**Soundness:** 3 good
**Presentation:** 3 good
**Contribution:** 3 good

**Summary:**

1.	In this paper, the authors study two popular bilevel optimizers AID-BiO and ITD-BiO, whose implementations involve different choices of loops. The authors first establish unified convergence analyses that are applicable to all implementation choices of loops for both optimizers.
2.	The authors specialize their results to characterize the computational complexities for all implementations, and they then provide an explicit comparison across different implementations.


**Questions:**

1.	What are the exact definitions of the notations $\Theta$ and $\Omega$?
2.	The authors may want to provide references on the hypergradient in AID-BiO to help readers to understand the algorithms.
3.	The authors consider a hyperparameter optimization problem on MNIST in the experiments on AID-BiO (in Line 315), while you consider another hyper-representation problem in the experiments on ITD-BiO (in Line 332). Why do you consider different problems for the two optimizers?


**Limitations:**

Yes, the authors adequately addressed the limitations and potential negative societal impact of their work.

**Strengths And Weaknesses:**

Strengths:
1.	The authors first establish unified convergence analyses that are applicable to all implementation choices of loops for both AID-BiO and ITD-BiO.
2.	The authors compare the computational complexities among all implementations of the optimizers. The numerical experiments further demonstrate their theoretical results.
3.	The paper is easy to follow.

Weaknesses:
1.	The exact definitions of some important notations such as $\Theta$ and $\Omega$ are missing.
2.	The authors consider a hyperparameter optimization problem on MNIST in the experiments on AID-BiO (in Line 315), while they consider another hyper-representation problem in the experiments on ITD-BiO (in Line 332). They may want to explain why they consider different problems for the two optimizers.
3.	In this paper, the authors only report the losses v.s. running time and the losses v.s. the number of iterations in experiments on AID-BiO and ITD-BiO, respectively. It would be more convincing if they report MV($\epsilon$)(the total number of Jacobian- and Hessian-vector product computations), Gc($\epsilon$)(the total number of gradient computations) or some more metrics in the experiments to support their theoretical results on the computational complexities.

---

> ### Author Response · Authors · 2022-08-02
> **Response to reviewer rZSp**
>
> Many thanks for providing the helpful review. In the revised version, we have made the changes based on the reviewer’s comments. All the changes are highlighted by the blue-colored texts.
>
> Q1: What are the exact definitions of the notations $\Theta$ and $\Omega$?
>
> A: Many thanks! We use $a(x)=\Theta(b(x))$ if  $cb(x)<a(x)<Cb(x)$ and $a(x)=\Omega(b(x))$ if  $a(x)>cb(x)$, where $c,C$ are universal constants. We have added the definitions in the revised pdf.
>
> Q2: The authors may want to provide references on the hypergradient in AID-BiO to help readers to understand the algorithms.
>
> A: Good point! We have added the following references [1,2] for understanding the hypergradient in AID-BiO.
>
> [1] Grazzi, R., Franceschi, L., Pontil, M., and Salzo, S. On the iteration complexity of hypergradient computation. In Proc. International Conference on Machine Learning (ICML), 2020.
>
> [2] Pedregosa, F. Hyperparameter optimization with approxi- mate gradient. In International Conference on Machine Learning (ICML), pp. 737–746, 2016.
>
> Q3: The authors consider a hyperparameter optimization problem on MNIST in the experiments on AID-BiO (in Line 315), while you consider another hyper-representation problem in the experiments on ITD-BiO (in Line 332). Why do you consider different problems for the two optimizers?
>
> A: Many thanks for the question! We have added the other experiment for each optimizer. Specifically, for ITD-BiO, we have added a plot (Fig. 3 in Appendix F) on the hyperparameter optimization problem on MNIST in the revision with $N=1$ and $N=20$, where it can be seen that $N=20$ achieves a lower error and hence our theory is validated. For AID-BiO, we have also added a plot (Fig. 4 in Appendix G) on the representation problem, and a conclusion similar to Fig. 1 can be observed. Both new experiments are in consistency with our theory.
>
> Q4: In this paper, the authors only report the losses v.s. running time and the losses v.s. the number of iterations in experiments on AID-BiO and ITD-BiO, respectively. It would be more convincing if they report $MV(\epsilon)$ (the total number of Jacobian- and Hessian-vector product computations), $Gc(\epsilon)$ (the total number of gradient computations) or some more metrics in the experiments to support their theoretical results on the computational complexities.
>
> A: Great suggestion! For AID-BiO,  since each matrix-vector (MV) computation takes the almost same time,  our running time curve takes a trend very similar to that of  the $MV(\epsilon)$ (similarly for $GC(\epsilon)$). To see this, we have added a new Fig. 4 in Appendix G in the revised pdf, which plots losses versus $MV$ (number of matrix-vector products) and $GC$ (number of gradients). It can be seen that these empirical results are also in consistency with our theoretical results.
>
> For ITD-BiO, the goal of our experiment is to show that No-loop ITD-BiO with $N=1$ induces a larger convergence error than $N$-$N$-loop ITD-BiO with $N=1$. In other words, we compare their losses after they converge, i.e., after $500$ iterations. Therefore, using the iteration as a metric serves the purpose of this comparison.

---

> > ### Comment · Reviewer_rZSp · 2022-08-07
> > **Thanks for the authors' response.**
> >
> > Thanks for the authors' rebuttal and their efforts to improve this work. The response has addressed my questions. I have raised my score to 7. Best wishes.

---

> > > ### Author Response · Authors · 2022-08-08
> > > **Thanks for the feedback!**
> > >
> > > We thank the reviewer very much for recognizing our efforts and for increasing the rating!
> > >
> > > Best, authors

---

### Author Response · Authors · 2022-08-09
**Thank all reviewers and ACs after rebuttal**

We first truly thank all reviewers’ insightful and constructive suggestions, which helped to significantly improve our paper! We also thank the area chair very much for great efforts into handling our paper during the review process!

Unfortunately, we regret that we have not received any response from Reviewer **KmZV** during this discussion period. Despite the absence of discussion engagement, we humbly believe that our detailed pointwise responses have clarified Reviewer **KmZV**’s questions. In particular, Reviewer **KmZV** had three major comments. **(1)** The submission is a summary of previous convergence results in [17]. In our response, we have elaborated in detail our new contributions beyond [17] in terms of new technical developments, much broader parameter regimes, relaxed assumptions, result generalizability, etc. **(2)** Comparison among the upper bounds of single- and double-loop algorithms rather than lower bounds. The comparisons of upper bounds (i.e., comparison of different $Q$ and comparison of different $N$ under $Q=1$) are new to the bilevel literature, which we believe are important steps in understanding their algorithmic designs. Such a type of comparisons have been widely used in optimization literature. **(3)** Choosing $N,Q$ as $1,20$ does not separate single- and double-loop and plot MV and GC. In our revision, we have added a new experiment in Fig. 2 with $N,Q$ chosen from $\{1,50\}$, and a new experiment in Fig. 4 comparing in MV and GC, both of which are consistent with our theoretical results.

We thank all reviewers’ and ACs' time and efforts again!

---

### Meta-Review · Area_Chair_jhhm · 2022-08-26

**Recommendation:** Accept
**Confidence:** Certain

**Metareview:**

There seems to be a clear consensus among reviewers about the paper being well written and addressing relevant research questions pertaining to bi-level optimization, with a particular focus on two popular bilevel optimizers AID-BiO and ITD-BiO. Furthermore, Reviewer Kzzc stressed that this work provides several interesting convergence results, with a practical echo in applications such as meta-learning, NAS, some HO problems, etc. Kzzc also pointed out that, the convergence for ITD has not been well studied, and the results on upper and lower bounds in this work can be a good complement. rZSp joined Kzzc by pointing out that the authors first establish unified convergence analyses that are applicable to all implementation choices of loops for both AID-BiO and ITD-BiO. While at first critical on some aspects of the work (notation, references, tests), Reviewer rZSp went on with a score increase following the constructive discussion with the authors. The most critical Reviewer was KmZV, who questioned the originality of contributions compared to [17]. While the authors did give a detailed response during the discussion, this reviewer did not react. Overall, based on the reviews and the discussion, I assess the paper to be a valuable addition tot he existing literature and recommend it to be accepted to NeurIPS 2022.

**Award:**

No

---

### Decision · Program_Chairs · 2022-09-14

Accept